https://doi.org/10.1038/s42003-019-0645-6　　**OPEN**

# A common mechanism allows selective targeting of GluN2B subunit-containing *N*-methyl-D-aspartate receptors

Julian A. Schreiber[1], Dirk Schepmann[1], Bastian Frehland[1], Simone Thum[1], Maia Datunashvili[2], Thomas Budde[2,3], Michael Hollmann[4], Nathalie Strutz-Seebohm[5], Bernhard Wünsch ![ORCID][1,3,6]* & Guiscard Seebohm[5,6]*

*N*-methyl-D-aspartate receptors (NMDARs), especially GluN2B-containing NMDARs, are associated with neurodegenerative diseases like Parkinson, Alzheimer and Huntington based on their high $Ca^{2+}$ conductivity. Overactivation leads to high intracellular $Ca^{2+}$ concentrations and cell death rendering GluN2B-selective inhibitors as promising drug candidates. Ifenprodil represents the first highly potent prototypical, subtype-selective inhibitor of GluN2B-containing NMDARs. However, activity of ifenprodil on serotonergic, adrenergic and sigma receptors limits its therapeutic use. Structural reorganization of the ifenprodil scaffold to obtain 3-benzazepines retained inhibitory GluN2B activity but decreased the affinity at the mentioned non-NMDARs. While scaffold optimization improves the selectivity, the molecular inhibitory mechanism of these compounds is still not known. Here, we show a common inhibitory mechanism of ifenprodil and the related 3-benzazepines by mutational modifications of the receptor binding site, chemical modifications of the 3-benzazepine scaffold and subsequent in silico simulation of the inhibitory mechanism.

[1] Institute of Pharmaceutical and Medicinal Chemistry, University of Münster, Corrensstr. 48, D-48149 Münster, Germany. [2] Institute of Physiology I, University of Münster, Robert-Koch-Str. 27a, D-48149 Münster, Germany. [3] Cells-in-Motion Cluster of Excellence (EXC 1003 - CiM), University Münster, Münster, Germany. [4] Department of Biochemistry I - Receptor Biochemistry, Ruhr University Bochum, Universitätsstr. 150, D-44801 Bochum, Germany. [5] Cellular Electrophysiology and Molecular Biology, Institute for Genetics of Heart Diseases (IfGH), Department of Cardiovascular Medicine, University Hospital Münster, Robert-Koch-Str. 45, D-48149 Münster, Germany. [6] These authors contributed equally: Bernhard Wünsch, Guiscard Seebohm. *email: wuensch@uni-muenster.de; guiscard.seebohm@ukmuenster.de

N-methyl-D-aspartate receptors (NMDARs) are members of the ionotropic glutamate receptor family and play important roles in a vast number of neurophysiological processes like memory, learning and synaptic plasticity[1–3]. Four out of seven different subunits (GluN1, GluN2A–D and GluN3A–B) form various heterotetrameric NMDARs[4,5]. Whereas two GluN1 subunits, which can be expressed in 8 different splice variants (GluN1–1a to GluN1–4b), are mandatory, the two remaining subunits are flexible combinations of the same or different GluN2 or GluN3 subunits leading to a large number of di- or triheteromeric receptors[6–9]. The receptor function mainly depends on the expressed GluN2 subunits[10]. Moreover, GluN2 subunits differ in tissue expression, localization and association with different pathophysiological mechanisms and diseases[11]. Especially, the GluN2B subunit is associated with neurodegenerative processes found in patients with Parkinson's, Alzheimer's and Huntington's Disease[12]. Therefore, selective modulators of NMDARs containing GluN2B subunits represent promising tools for further investigation and potential treatment of these diseases.

All subunits share a common architecture consisting of four different domains. The carboxy-terminal domain (CTD) on the intracellular side is responsible for regulation and receptor trafficking by phosphorylation or binding of secondary messengers[13]. The ion pore is formed by the helices M1, M3, M4 and the re-entry loop M2 of the transmembrane domain (TMD)[14]. In resting state, the ion flux through the pore is blocked by $Mg^{2+}$[15]. For receptor activation binding of both endogenous ligands glycine and (S)-glutamate at the ligand binding domain (LBD) and removal of $Mg^{2+}$ ion block by slight depolarization of the embedding membrane is required[16]. Under physiological conditions receptor activation causes an efflux of $K^+$ and an influx of $Na^+$ and $Ca^{2+}$ ions along their concentration gradients resulting in activation of $Ca^{2+}$ mediated secondary processes and membrane depolarization[17,18]. As a consequence of $Ca^{2+}$ permeability overactivation of NMDARs causes excitotoxicity and subsequent cell death, which explains the role of NMDARs in neurodegenerative diseases[19].

The extracellular LBD of all GluN subunits forms a clamp-shell like structure. While the LBD of GluN2 subunits is responsible for (S)-glutamate binding, GluN1 and GluN3 subunits bear binding sites for glycine. Binding of both agonists results in clamp-shell closing and subsequently to ion channel opening[20]. In contrast to AMPA and kainate receptors, NMDAR channel opening is modulated by the extracellular amino-terminal domain (ATD), which shows sequence and structure homology with the LBD[14,21]. Similar to the LBD, the ATD has a bilobed structure, which can be subdivided into the R1 and R2 subdomains[22]. Cryo-EM analysis and different crystal structures of the complete receptor revealed distinct conformational changes of the ATD influencing the structure of the LBD[14,22].

The influence of ATD on gating mechanism is underlined by a variety of different NMDAR modulators binding to the ATD such as $Zn^{2+}$, $H^+$, polyamines and phenylethanolamines[23–25]. The prototype of the phenylethanolamine-based GluN2B-selective NMDAR modulators is ifenprodil, which was originally designed as an $\alpha_1$-adrenoceptor antagonist[26]. Ifenprodil shows high selectivity for GluN2B-containing NMDARs and represents the starting point for the development of more GluN2B subunit selective NMDAR modulators such as CP-101,606 and Ro 25-6981[27,28]. Although ifenprodil reveals high selectivity for GluN2B subunit-containing NMDARs, its interaction with adrenergic, serotonergic and σ receptors limit its therapeutic use as a neuroprotective agent. One reason for the activity at these receptors could be the high flexibility of the ifenprodil scaffold allowing adaptation to other binding sites as well.

The ifenprodil binding site was confirmed by crystallization with the isolated ATD and the complete receptor[22,29]. It is located at the interface within the ATD between the GluN1 and GluN2B subunit and connects the GluN2B R1 subdomain with the GluN2B R2 subdomain. As a result of binding site location GluN1 and GluN2B residues are involved in the binding process. Electrophysiological studies and subsequent mathematical state modeling of the binding process suggest an induced-fit mechanism, which stabilize the closed state and restricts the transition to the open state[30]. In addition, cryo-EM studies with the ifenprodil derived compound Ro 25-6981 reveal crucial conformational changes in the ATD and LBD, which are comparable to previous published cryo-EM studies of different NMDA receptor states[31].

Although the binding site and some specific interactions with the receptor are known, the molecular mechanism of ifenprodil and related compounds is not completely understood. To investigate the mechanism, we used compound rac-1 with a conformationally more restricted 3-benzazepine scaffold. rac-1 is in agreement with the pharmacophore model of ifenprodil derived compounds bearing two aromatic systems connected by a linker with a central basic amine[32]. Previous studies of rac-1 indicated that restriction of the molecular flexibility increased selectivity over adrenergic and σ receptors without losing activity at GluN2B-containing NMDARs, which is a critical benefit compared to ifenprodil[33]. Moreover, the enantiomers of rac-1 were separated and (R)-1 was identified as eutomer[34].

In this study, ifenprodil, 1 and related 3-benzazepines are used to determine the inhibitory mechanism of these negative allosteric modulators (NAMs) by well-aimed modifications of the ligands, mutations of the ligand binding site and in silico simulations of the molecular mechanism. Our results demonstrate a common inhibition mechanism for ifenprodil and 3-benzazepines, which is mainly generated by an aromatic interaction with GluN2B F176 preventing reorientation of the α5 helix that is prerequisite for receptor opening.

## Results

**3-Benzazepines exert high selectivity and similar activity.** 3-Benzazepines 1 and 2 exhibit reduced flexibility in the phenylethanolamine portion, but leave distances between H-bond donors, H-bond acceptors and the two aromatic rings unchanged, which resulted in reduced affinity to non-NMDARs (Fig. 1a–d)[33]. Furthermore, 3-benzazepines 3–7 with more pronounced structural modifications were designed and pharmacologically evaluated (Fig. 1e–j). In receptor binding studies with [³H]ifenprodil 3-benzazepines 1–7 showed GluN2B affinity in the low to medium nanomolar range (Table 1) indicating strong interactions with the ifenprodil binding site.

In order to test, whether this high GluN2B affinity translates into high antagonistic activity, 3-benzazepines (R)-1, (S)-1, 2 and ifenprodil were investigated in TEVC experiments using oocytes expressing GluN1-1a/GluN2B (Fig. 2a–f, Table 1). While (S)-1 shows comparable activity to ifenprodil, (R)-1 and 2 display mildly increased activity confirming a preference for 3-benzazepines with (R)-configuration in benzylic position. A similar tendency was already observed in previous studies[34].

Beside a significantly reduced affinity to non-NMDARs, the selectivity for NMDARs containing the GluN2B subunit is an important property for ifenprodil-derived compounds. Therefore, the activity of 10 μM (R)-1, (S)-1 and 2 was evaluated by TEVC measurements at GluN2A-, GluN2C- and GluN2D-containing NMDARs and compared to the activity of 10 μM ifenprodil (Supplementary Table 1). The results do not show significant

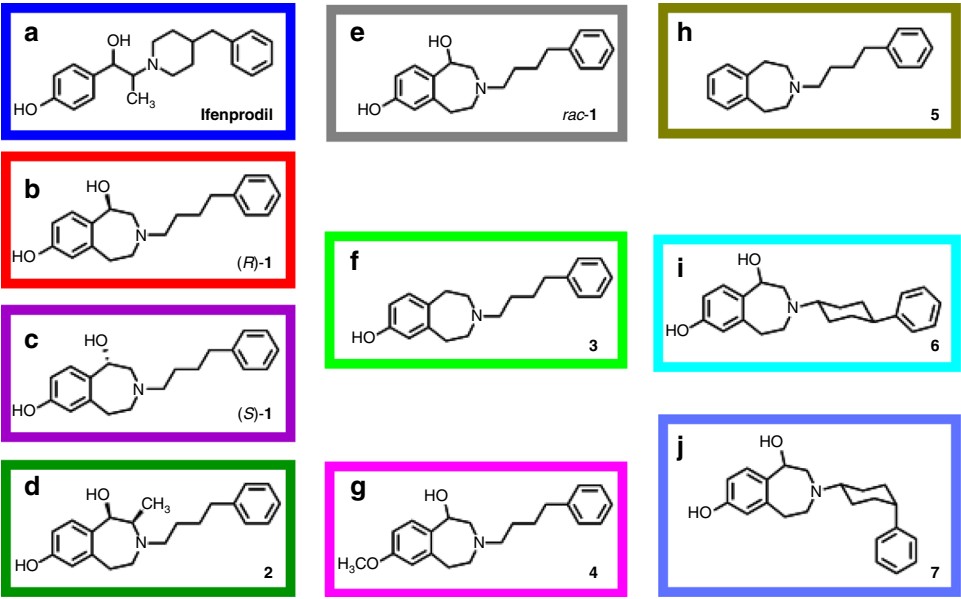

**Fig. 1** Structures of investigated 3-benzazepines and ifenprodil. **a–d**. Structures of the highly active compounds ifenprodil (**a**), (*R*)-**1** (**b**), (*S*)-**1** (**c**) and **2** (**d**). Border color serves as an identifier for the compounds in Figs. 2, 4, 5, 7. **e–j**. Structures of 3-benzazepines *rac*-**1** (**e**), **3** (**f**), **4** (**g**), **5** (**h**), **6** (**i**) and **7** (**j**) with more pronounced structural modifications. Border color serves as an identifier for the compounds in Fig. 4

### Table 1 Affinity and activity of ifenprodil and 3-benzazepines 1–7

| Compound | affinity | | TEVC activity | | |
|---|---|---|---|---|---|
| | $K_i$ + SEM [nM] | $IC_{50}$ ± SE [nM] | $A_2$ ± SE [%] | $p$ | $n$ |
| ifenprodil | 10 ± 1[b] | 264 ± 27 | 94 ± 2 | 0.97 | 24 |
| *rac*-**1** | 84 ± 18[a] | 94 ± 8 | 93 ± 2 | 1.25 | 24 |
| (*R*)-**1** | 30 ± 8[a] | 53 ± 3 | 93 ± 1 | 1.11 | 24 |
| (*S*)-**1** | 740 ± 61[a] | 206 ± 23 | 95 ± 2 | 0.98 | 24 |
| **2** | 111 ± 53[b] | 91 ± 7 | 97 ± 2 | 1.08 | 24 |
| **3** | 10 ± 3[b] | 3367 ± 926 | 92 ± 8 | 0.93 | 18 |
| **4** | 681 ± 133[b] | 1305 ± 483 | 100 ± 9 | 0.70 | 18 |
| **5** | 225 ± 3[b] | 9 ± 2%[c] | – | – | 3 |
| **6** | 29 ± 2[b] | 2629 ± 296 | 93[d] | 0.70 | 12 |
| **7** | 39 ± 6[b] | 7211 ± 1774 | 93[d] | 0.62 | 12 |

$K_i$ values were determined by displacement of [³H]ifenprodil in three independent experiments. Activity was determined by TEVC at holding potential of −70 mV using GluN1-1a/GluN2B expressing oocytes and is given as $IC_{50}$ value. $A_2$ represents the maximum inhibition of compound derived from dose-response curve, while $p$ represents the slope of the dose-response curve. $n$ represents the number of independent oocytes, which were used to generate the dose-response curve
[a]$K_i$ values were previously published[34]
[b]Newly recorded $K_i$ values
[c]Inhibition (% ± SEM) of ion flux at a compound concentration of 30 µM
[d]Due to non-saturating inhibition, $A_2$ was set as identical with *rac*-**1**

differences in activity of (*R*)-**1** and (*S*)-**1** compared to the activity of ifenprodil confirming that the 3-benzazepine scaffold does not decrease selectivity against NMDARs with different GluN2 subunits. On the other hand, the additional methyl group in 2-position of compound **2** increased slightly the activity at GluN2C and GluN2D subunit-containing NMDARs compared to ifenprodil. 10 µM of **2** led to ion current inhibitions of 56 ± 2% (SEM; GluN2C) and 55 ± 2% (SEM; GluN2D). Altogether, (*R*)-**1** shows a superior pharmacological profile, since it has in comparison to ifenprodil slightly increased activity at GluN2B subunit-containing NMDARs, decreased affinity at non-NMDARs and similar selectivity over NMDARs with different GluN2 subunits.

To examine the properties of (*R*)-**1** in a more physiological system, the inhibitory activity of (*R*)-**1** was evaluated by whole cell patch-clamp recordings in hippocampal slices. Evoked excitatory postsynaptic currents (EPSCs) were measured in presence and absence of the compound in CA1 pyramidal neurons, which receive excitatory glutamatergic inputs from Schaffer collaterals originating in the CA3 region. The NMDAR-dependent component of the EPSC was isolated pharmacologically (see "Materials and methods") and the effect of (*R*)-**1** was compared to the effect of ifenprodil under similar conditions. After recording of a stable EPSC baseline for 5 min, slices were washed with different concentrations of ifenprodil and (*R*)-**1** while EPSCs were constantly recorded (Fig. 2g–j). Application of 1 µM and 6 µM ifenprodil reduced the EPSC amplitudes significantly to 74.4 ± 8.9% (SEM, $p < 0.05$, $n = 5$) and 22.7 ± 4.7% (SEM, $p < 0.001$, $n = 5$) of the initial baseline, respectively. Washout of ifenprodil revealed a partial recovery of the EPSC during the observed recording period to an EPSC amplitude of 31.9 ± 5% (SEM, $p > 0.05$, $n = 5$). In the presence of 2 µM and 7 µM (*R*)-**1**, EPSC amplitudes were reduced to 60.2 ± 2.8 % (SEM, $p < 0.001$, $n = 5$) and 13.8 ± 1.4% (SEM, $p < 0.001$, $n = 5$) of the initial baseline value, respectively. Washout of (*R*)-**1** caused a significant recovery of EPSC amplitudes during the observed recording period to 30.9 ± 3.6% (SEM, $p < 0.01$, $n = 5$). These data identify (*R*)-**1** as a possible candidate for a new generation of ifenprodil derived compounds. To uncover the inhibitory mechanism, we focused on the examination of critical interactions within the binding site.

**Docking reveal five common interaction zones IZ1–IZ5**. Since 3-benzazepines can displace [³H]ifenprodil from its binding site and are also able to inhibit the GluN2B-containing NMDARs, a common binding site and a common inhibitory mechanism can be assumed. To identify critical molecular interactions at the binding site, docking was performed with 3-benzazepines **1–7**. The highly active compounds ifenprodil, (*R*)-**1**, (*S*)-**1**, and **2** show similar docking conformations and interactions with the identical amino acids (Fig. 3a–d), while docking results of **3–7** differ remarkably (Supplementary Fig. 1).

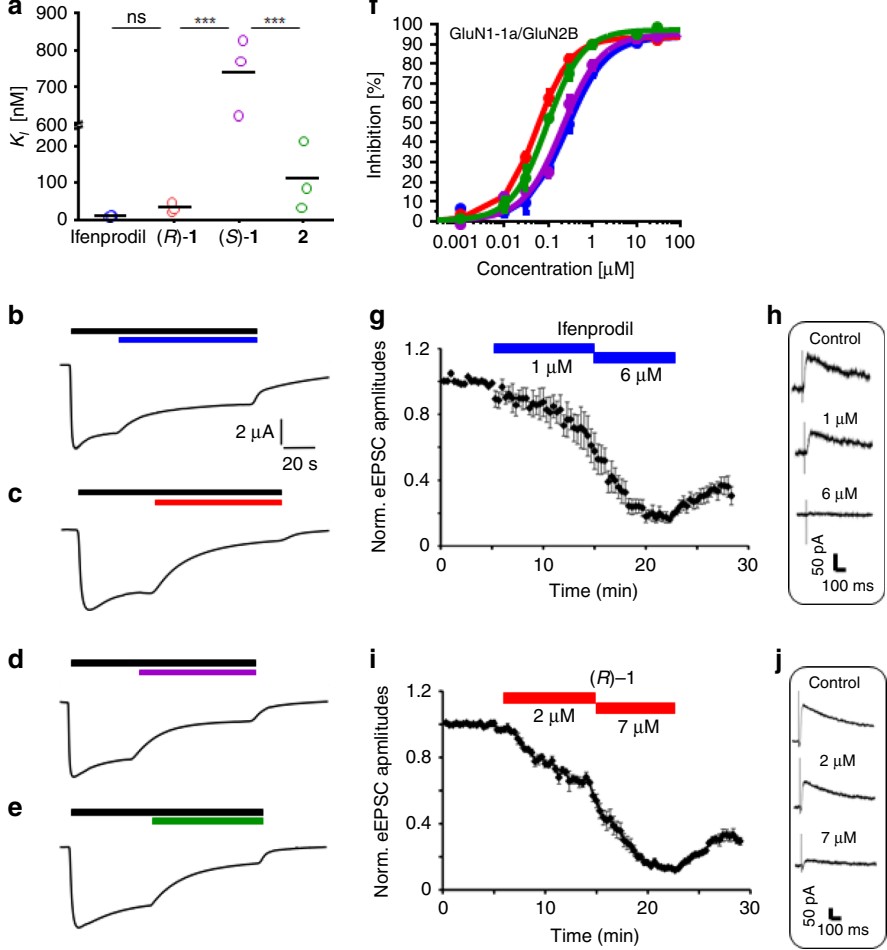

**Fig. 2 Affinity and activity of ifenprodil and 3-benzazepines (R)-1, (S)-1 and 2. a.** $K_i$ values of ifenprodil, (R)-1, (S)-1 and 2 presented as single data points (n = 3 for (R)-1, (S)-1 and 2 and n = 6 for ifenprodil) and mean (black bar). **b–e** Representative current traces of GluN1-1a/GluN2B expressing oocytes activated by 10 μM glycine and 10 μM (S)-glutamate (black bar), inhibited by 300 nM compound (colored bars: ifenprodil (blue), (R)-1 (red), (S)-1 (purple), 2 (green)). **f** Dose-response curves resulting from percental inhibition ± SEM of ifenprodil, (R)-1, (S)-1 and 2 inhibiting GluN1-1a/GluN2B expressing oocytes at holding potential of -70 mV activated by 10 μM glycine and 10 μM (S)-glutamate (color code is defined in Fig. 1a–d). Dose-response curves were generated for each compound from recordings of n = 24 independent oocytes (see Table 1). **g** Averaged time course of normalized EPSC current amplitudes in CA1 pyramidal neurons (n = 5 independent experiments) showing the concentration-dependent effect of 1 μM and 6 μM ifenprodil and partial wash out. **h** Representative traces of evoked NMDA receptor-dependent EPSCs during baseline and following 1 μM and 6 μM ifenprodil application. **i** Averaged time course of normalized EPSC current amplitudes in CA1 pyramidal neurons (n = 5 independent experiments) showing the concentration-dependent effect of 2 μM and 7 μM (R)-1 and partial wash out. **j** Representative traces of evoked NMDA receptor-dependent EPSCs during baseline and following 2 μM and 7 μM (R)-1 application

Amino acids most frequently observed as interaction partners for ifenprodil and 3-benzazepines 1–7 during docking studies were classified into five interaction zones IZ1–IZ5, which are illustrated exemplarily for (R)-1 (Fig. 3e, f). IZ1 consists of deprotonated GluN2B E236 forming an H-bond interaction with the phenolic OH-group of the ligands. This interaction was also reported in previous studies[32]. IZ2 is formed by GluN1-1a L135 and GluN1-1a S132 resulting in hydrophobic and H-bond interactions with the 3-benzazepine scaffold and its benzylic OH moiety. While IZ4 shows hydrophobic (GluN1-1a F113, GluN2B I111) and H-bond interactions (GluN2B Q110), IZ3 and IZ5 consist predominantly of aromatic interactions with GluN2B F114 and GluN2B F176. In order to analyze the contribution of the ligand interactions with the different interaction zones to their inhibitory activity, mutants of IZ1–IZ5 were generated and the impact on ifenprodil, (R)-1, (S)-1 and 2 activity was evaluated by TEVC. $IC_{50}$ values for the inhibition of

mutated GluN1-1a/GluN2B receptors were used to calculate the $IC_{50}$ shift factor, which compares the inhibition of wildtype and mutated GluN1-1a/GluN2B NMDARs (see formula 3 in data analysis part). $IC_{50}$ values and fit parameters for all mutants and compounds are summarized in Supplementary Tables 2–5.

**IZ1 and IZ2 are defined by compound substituents.** In IZ1 and IZ2 the compounds interact with the phenol and benzazepine part by H-bond and hydrophobic interactions (Fig. 4a). Whilst IZ1 is defined by an interaction with the GluN2B subunit, residues of IZ2 are located at the GluN1-1a subunit. Docking suggests H-bonding between the benzylic OH-group of the compounds and the backbone of GluN1-1a S132. Since backbone interactions are not analyzable by mutagenesis, 3-benzazepine 3 without benzylic OH-group was synthesized to evaluate the importance of this interaction. Recording the inhibitory activity

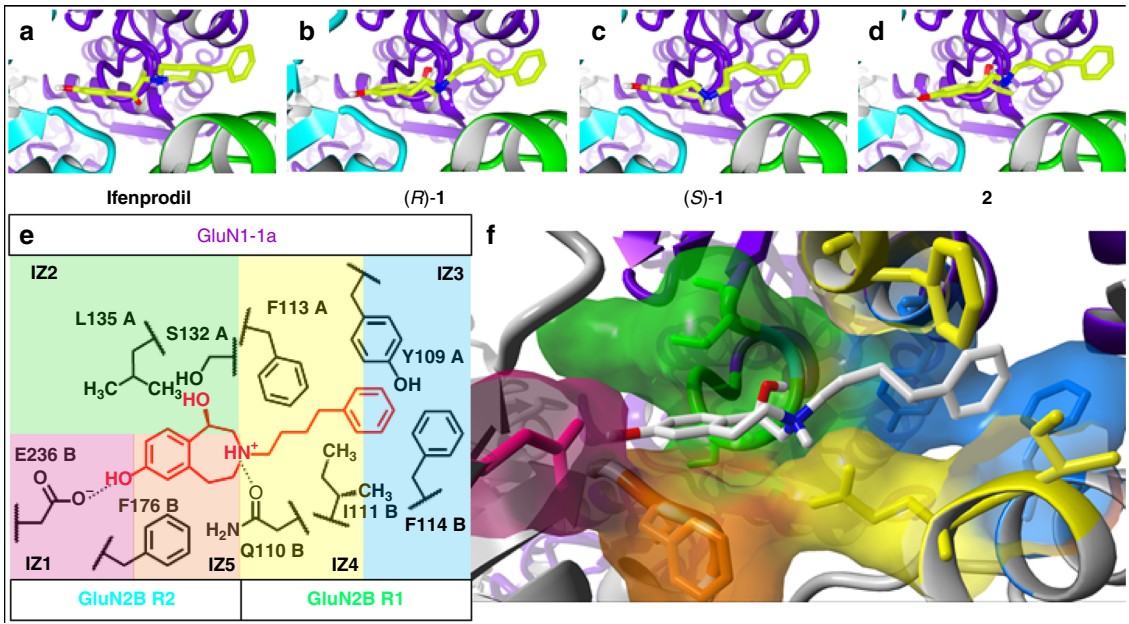

**Fig. 3** Docking studies result in five common zones of interaction. **a–d** Docking poses of ifenprodil (**a**), (*R*)-**1** (**b**), (*S*)-**1** (**c**) and **2** (**d**) (CPK color code with green for C-atoms) at crystal structure of GluN1-1a (purple)/GluN2B NMDAR with colored subdomains R1 (green) and R2 (cyan) (PDB 4PE5). **e** Schematic illustration of defined interaction zones IZ1–IZ5 with associated residues and bound (*R*)-**1** (red). Residues from GluN1-1a marked with A, GluN2B residues with B. IZ1, IZ2 and IZ4 show hydrophobic and H-bond interactions, while IZ3 and IZ5 consist of aromatic interactions. IZ1 and IZ5 are located at GluN2B R2 subdomain, while IZ3 and IZ4 are built from residues of GluN1 ATD and GluN2B R1. **f** Graphical presentation of interaction zones IZ1 to IZ5 and associated residues at ifenprodil binding site of GluN1-1a/GluN2B receptor (PDB 4PE5) with docked (*R*)-**1** (CPK color code, C-atoms in gray). Residues of interaction zones are colored in pink (IZ1), green (IZ2), blue (IZ3), yellow (IZ4), and orange (IZ5)

using the mutant GluN1-1a S132A showed only small shifts in $IC_{50}$ values for the four most active compounds ($IC_{50}$ shift factors 2.0 to 4.3) and no considerable reduction in maximal inhibition (Fig. 4b). However, the $IC_{50}$ value of **3** without benzylic OH moiety was 36-fold increased compared to *rac*-**1** indicating the importance of this OH group for high activity (Fig. 4c).

To evaluate the relevance of hydrophobic interactions with GluN1-1a L135, the hydrophobic isobutyl side chain of leucine was exchanged by the more hydrophilic carbamoylmethyl side chain of asparagine. This modification of the GluN1-1a subunit resulted in moderate to strong shifts of $IC_{50}$ values ($IC_{50}$ shift factors 5.5. to 17.8) and reduced maximum inhibition for all tested compounds except methylated analog **2** (Fig. 4d).

The mutation GluN2B E236 to alanine in IZ1 resulted in inconsistent changes in $IC_{50}$ values. The inhibition of the ion channel by the methylated 3-benzazepine **2** was marginally affected by the mutation E236A. However, $IC_{50}$ values of (*R*)-**1**, (*S*)-**1** and ifenprodil were moderately to strongly increased ($IC_{50}$ shift factors 3.6 to 9.6; Fig. 4e). To clarify the importance of IZ1 for high inhibitory activity methyl ether **4** was included into this study and tested at GluN1-1a/GluN2B wildtype NMDAR expressing oocytes: **4** showed a 14-fold $IC_{50}$ shift compared to *rac*-**1** (Fig. 4c).

To further support the hypothesis that benzylic and phenolic OH groups contribute to the activity of the 3-benzazepines in an additive manner, 3-benzazepine **5** without both OH groups was included into this study and tested at GluN1-1a/GluN2B NMDARs. At a concentration of 30 μM, **5** was almost inactive (Fig. 4f, Table 1). This result corroborates the hypothesis of additive effects of both OH moieties on channel inhibition. Although inhibitory activities of **3**, **4**, and **5** being dramatically reduced, GluN2B affinity of **3**, **4**, and **5** in the [³H]ifenprodil competition assay is still in nanomolar range (Table 1). This observation underlines an uncoupled relationship for affinity and activity for the 3-benzazepines.

**Aromatic interactions in IZ3 highly influence the activity.** Docking experiments revealed clear interactions between the terminal phenyl moiety and a hydrophobic pocket formed by the aromatic side chains of GluN1-1a Y109 and GluN2B F114 (Fig. 4a). This hydrophobic pocket has been described as critical for ligand potency of different compounds like ifenprodil and 93-31[35]. To analyze the relevance of these π/π-interactions in detail, amino acids were mutated to serine (GluN1-1a Y109S) and alanine (GluN2B F114A). Whereas the mutation GluN1-1a Y109S had only minor influence on the activity ($IC_{50}$ shift factors 0.4-3.7), GluN2B F114A decreased the activity of ifenprodil, (*R*)-**1**, (*S*)-**1** and **2** considerably showing $IC_{50}$ shift factors between 7.3 and 199 (Fig. 4g, h). These results implicate a strong aromatic interaction with the GluN2B subunit, which is only weakly modulated by the GluN1-1a subunit.

In summary, it was hypothesized, that ifenprodil and related 3-benzazepines require simultaneous interactions with IZ1, IZ2, and IZ3 to achieve high inhibitory activity. To enable all postulated interactions, the ligands must adopt a defined conformation within the binding site. To confirm this hypothesis, diastereomeric phenylcyclohexyl-substituted 3-benzazepines **6** and **7** were included in this study. The cyclohexyl spacer leads to a defined distance and orientation of the terminal phenyl moiety in relation to the basic amine. Both phenylcyclohexyl derivatives **6** and **7** exhibit high GluN2B affinity, but rather low inhibitory activity (Fig. 4i, j). The fixed orientation of the terminal phenyl moiety disrupts the optimal positioning of the ligands to form interactions with IZ1–IZ3, which resulted in a significant loss of activity without losing affinity (Fig. 4i, j, Table 1, Supplementary Fig. 1I–L). This observation confirms the hypothesis that simultaneous interactions with IZ1–IZ3 are required for high activity. Furthermore, 3-benzazepines **6** and **7** represent a further example for uncoupled relationships between affinity and inhibitory activity.

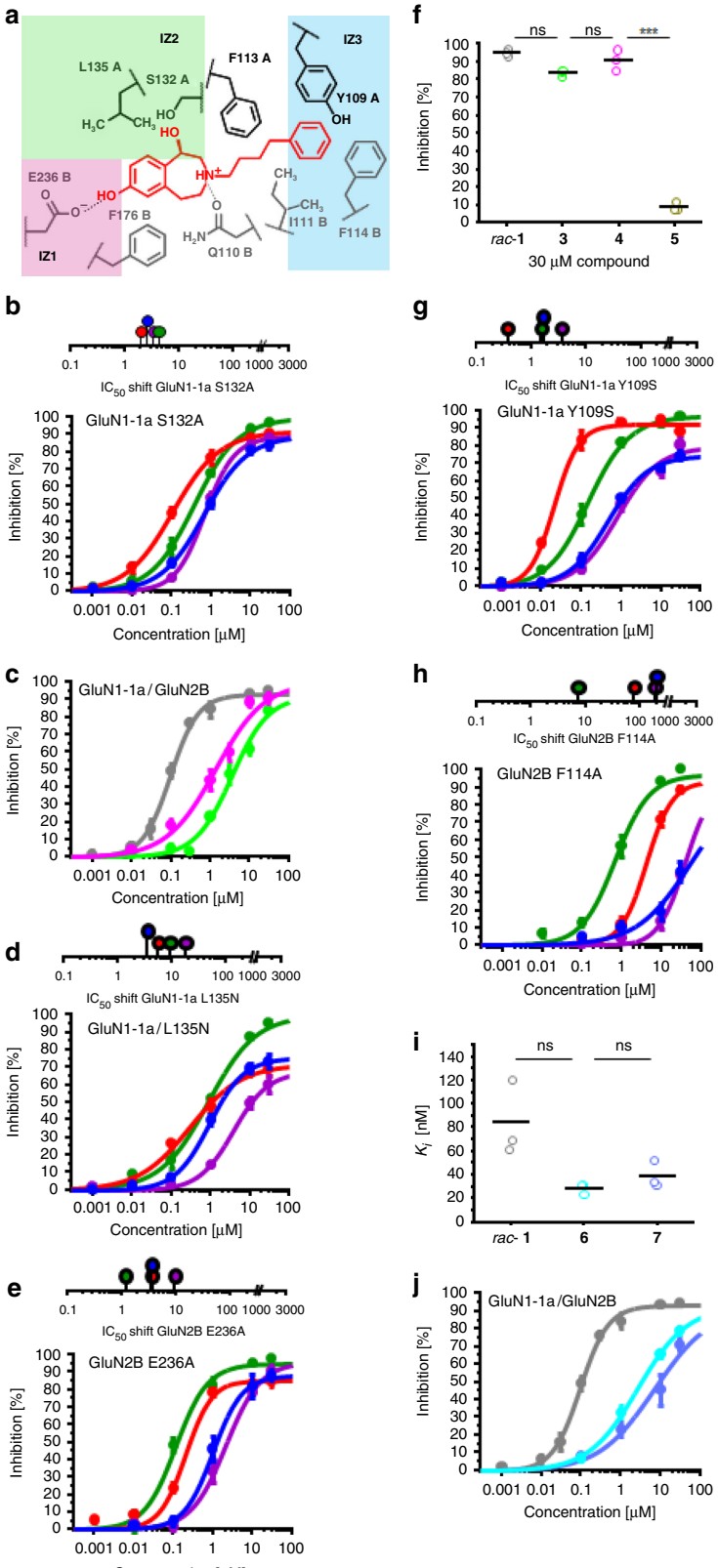

**High activity is not based on interactions within IZ4.** Although replacement of the flexible butyl spacer by a rigid cyclohexyl moiety proved the importance of a flexible positioning of the terminal phenyl moiety, the involvement of the compound spacer and the basic amine in direct interactions with surrounding amino acids remains unclear. Docking conformations predict hydrophobic and H-bond interactions with GluN1-1a F113, GluN2B Q110, and GluN2B I111 (Fig. 5a). In particular the H-bond between the side chain of GluN2B Q110 and the protonated basic amine was previously reported and postulated as an important interaction for the whole class of compounds[22,29]. It was speculated that interactions with GluN2B Q110 help to

Fig. 4 Interaction zones IZ1, IZ2, and IZ3. a Schematic depiction of interaction zones IZ1 (pink), IZ2 (green) and IZ3 (blue) and the corresponding H-bond, hydrophobic and aromatic interactions with (R)-1 (red). b IC₅₀ shift factors and dose-response curves resulting from percental inhibition ± SEM of ifenprodil (blue), (R)-1 (red), (S)-1 (purple) and 2 (green) at oocytes expressing mutated NMDARs GluN1-1a S132A. Curves were generated from recordings of n = 18 independent oocytes (see Supplementary Tables 2–5). c Dose-response curves resulting from percental inhibition ± SEM of rac-1 (gray), 3 (green) and 4 (pink) at GluN1-1a/GluN2B expressing oocytes. Curves were generated from recordings of n = 24 (rac-1) or n = 18 (3, 4) independent oocytes (see Table 1). d, e IC₅₀ shift factors and dose-response curves resulting from percental inhibition ± SEM of ifenprodil (blue), (R)-1 (red), (S)-1 (purple) and 2 (green) at oocytes expressing mutated NMDARs with GluN1-1a L135N (d) and GluN2B E236A (e). Curves were generated from recordings of n = 15–18 independent oocytes (see Supplementary Tables 2–5). f Inhibition data of 30 μM rac-1 (grey), 3 (green), 4 (pink), and 5 (brown) at oocytes expressing wildtype GluN1-1a/GluN2B NMDA receptors presented as single data points (n = 3 independent oocytes per compound) and mean (black bar). g, h IC₅₀ shift factors and dose-response curves resulting from percental inhibition ± SEM of ifenprodil (blue), (R)-1 (red), (S)-1 (purple), and 2 (green) using oocytes expressing mutated NMDARs with GluN1-1a Y109S (g) or GluN2B F114A (h). Curves were generated from recordings of n = 12–18 independent oocytes for each compound (see Supplementary Table 2–5). i Kᵢ values of rac-1, 6, and 7 presented as single data points (n = 3 independent experiments) and mean (black bar). j Dose-response curves resulting from percental inhibition ± SEM of rac-1 (grey), 6 (turquoise), and 7 (blue) at wildtype GluN1-1a/GluN2B NMDARs. Curves were generated from recordings of n = 24 (rac-1) or n = 12 (6, 7) independent oocytes (see Table 1)

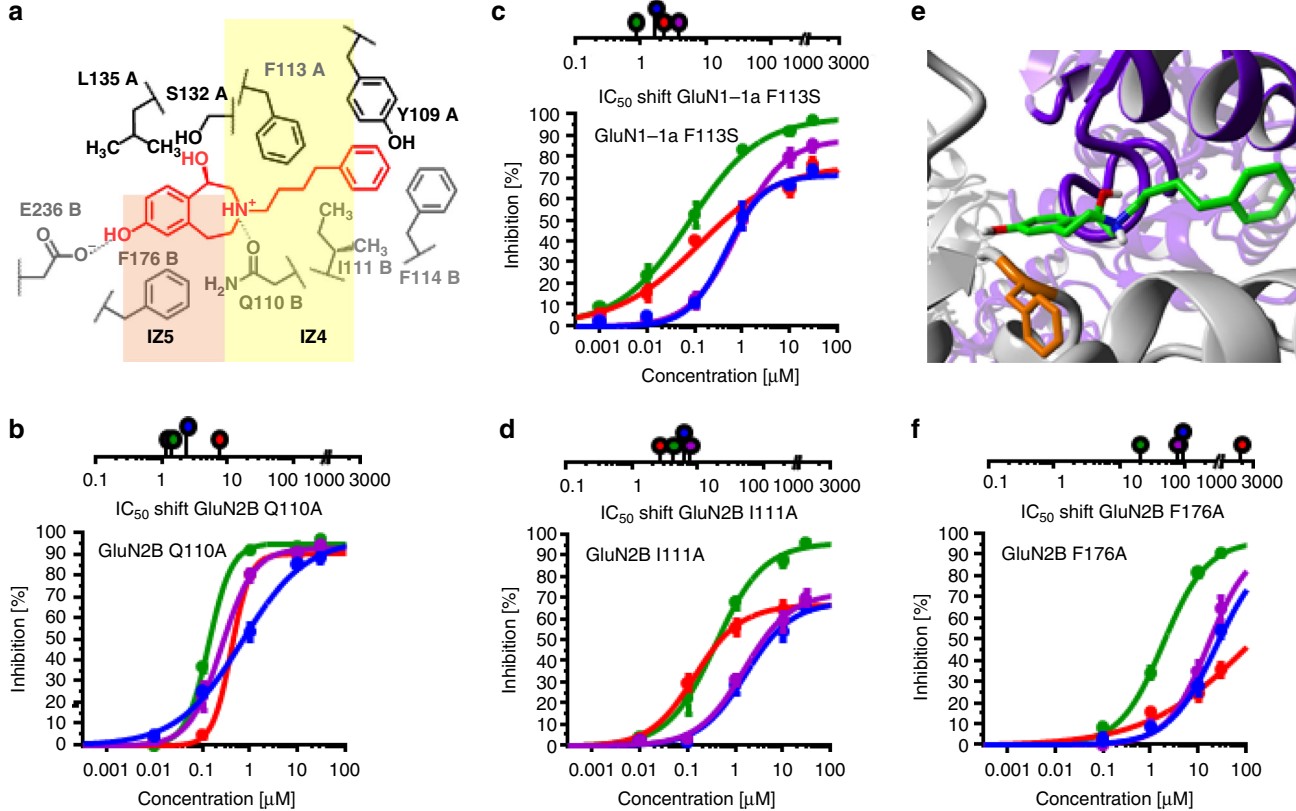

Fig. 5 Interaction zones IZ4 and IZ5. a Schematic depiction of interaction zones IZ4 and IZ5 displaying hydrophobic, aromatic and H-bond interactions with (R)-1 (red). b–d IC₅₀ shift factors and dose-response curves resulting from percental inhibition ± SEM for ifenprodil (blue), (R)-1 (red), (S)-1 (purple) and 2 (green) using oocytes expressing mutated GluN1-1a/GluN2B receptors with GluN2B Q110A (b), GluN1-1a F113S (c) and GluN2B I111A (d). Curves were generated from recordings of n = 15–18 independent oocytes for each compound (see Supplementary Tables 2–5). e Docking conformation of (R)-1 (CPK coloring with green for C-atoms) at ifenprodil binding site of GluN1-1a (purple)/GluN2B (gray) receptor with residue GluN2B F176 in orange. f IC₅₀ shift factors and dose-response curves resulting from percental inhibition ± SEM for ifenprodil (blue), (R)-1 (red), (S)-1 (purple), and 2 (green) using oocytes expressing mutated NMDARs with GluN2B F176A. Curves were generated from recordings of n = 12 independent oocytes for each compound (see Supplementary Tables 2–5)

stabilize the ligand orientation within the binding site[32]. However, mutation GluN2B Q110A did not lead to a pronounced reduction of activity for ifenprodil, (R)-1, (S)-1 and 2, concluding that this interaction is of minor importance for the inhibitory mechanism (Fig. 5b).

The hydrophobic interactions with GluN1-1a F113 and GluN2B I111 seem also to be of minor importance for high activity, since TEVC experiments with the mutants GluN1-1a F113S and GluN2B

I111A showed only weak to moderate shifts of IC₅₀ values (Fig. 5c, d). Only GluN2B I111A generated a notable reduction of maximal inhibition for all compounds except 2, which can be interpreted as a hydrophobic interaction between the amino acid side chain and the butyl spacer of these compounds. Nevertheless, the low relevance of the interactions with the spacer and the protonated amine underlines the hypothesis that high activity of 3-benzazepines is generated by interactions within the interaction zones IZ1–IZ3.

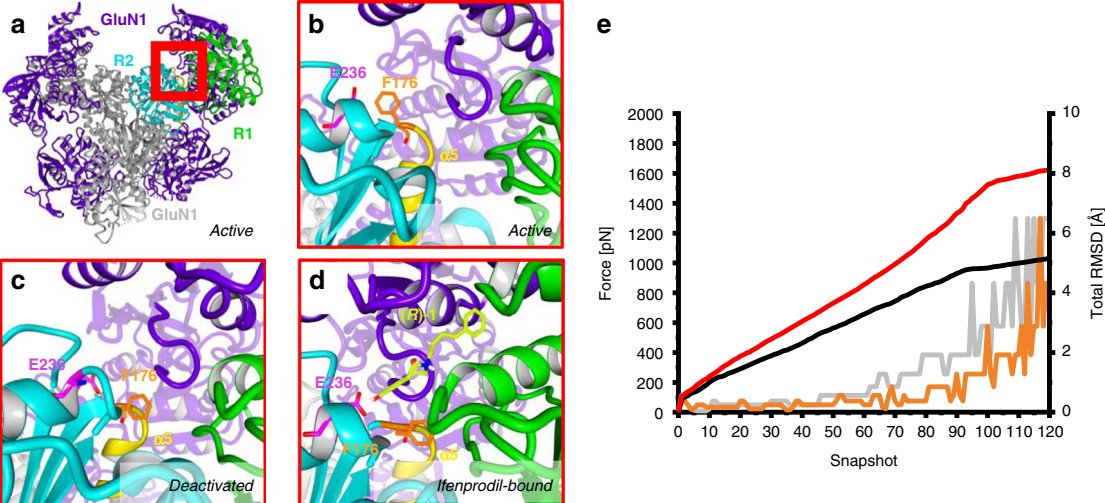

**Fig. 6 Homology models and morphing. a** Homology model of GluN1-1a (purple)/GluN2B (gray) ATD and LBD in active conformation. Subdomains of GluN2B ATD are colored in green (R1) and cyan (R2). Ifenprodil binding site is marked by red box. **b** Ifenprodil binding site in active conformation with vertical side chain orientation of GluN2B F176 (orange), H-bond acceptor GluN2B E236 (magenta) and α5 helix displaced in the direction towards ATD (yellow). **c** Ifenprodil binding site in deactivated conformation with horizontal side chain orientation for GluN2B F176 (orange), H-bond acceptor GluN2B E236 (magenta) and α5 helix displaced in the direction towards to LBD (yellow). **d** Ifenprodil binding site in ifenprodil-bound conformation with horizontal side chain orientation of GluN2B F176 (orange), H-bond acceptor GluN2B E236 (magenta) and docked ligand (*R*)-**1** (green). The α5 helix (yellow) shows higher displacement in the direction towards the LBD than in the deactivated state. **e** Morphing results: Total RMSD for transition of active to deactivated (black) and active to ifenprodil-bound state (red) indicates increased conformational changes by ifenprodil binding. The morphing force required for the transition from active to deactivated state (gray) is larger than the force for transition of active to ifenprodil-bound state (orange) suggesting that ifenprodil binding induces deactivated receptor conformation

**The key aromatic interaction is located in IZ5.** IZ5 is formed by the aromatic amino acid GluN2B F176 interacting via π/π-interaction with the phenolic part of ifenprodil and 3-benzazepines (Fig. 5e). To disrupt the postulated interaction, phenylalanine was mutated into alanine. This mutation led to the most distinct activity reduction with $IC_{50}$ shift factors of 20 to 1977 (Fig. 5f). Therefore, it is concluded that π/π-interactions between GluN2B F176 and the ligands contribute considerably to their inhibitory activity and are directly involved in the molecular mechanism of inhibition.

**Simulation of receptor transition and inhibitory mechanism.** After identifying the crucial interaction zones for high activity, a model for the receptor transition was developed to analyze the mechanism of inhibition in more detail. For this purpose, cryo-EM and crystal structures of the active (PDB 5FXG), the deactivated (PDB 5FXI) and the ifenprodil-bound state (PDB 4PE5) were used for morphing, a type of steered molecular dynamics simulation. Due to missing side chain information in the original structures and the insufficient resolution of the TMD in the active state structure, homology models, were generated using just sequences and structural information of the ATD and LBD (Fig. 6a). The process of morphing transforms the homology model of the active state (homology model of 5FXG) into the model of deactivated (homology model of 5FXI) or ifenprodil-bound state (homology model of 4PE5) by applying pulling forces to the structure until convergence of both homology models is reached. This in silico simulation allows to study the required steps during the transition from one state model to another. While cryo-EM structures could not provide detailed information about side chain orientation, the generated homology models revealed distinct differences in sidechain orientation of GluN2B F176 (Fig. 6b–d).

In the active state, the phenyl moiety of GluN2B F176 is oriented vertically to the receptor and the GluN2B

R2 subdomain is displaced towards the GluN1 ATD and GluN2B R1. During the transitions into the deactivated or ifenprodil-bound models, the entire GluN2B R2 subdomain including residue GluN2B F176 undergoes pronounced repositioning. The displacement of GluN2B F176 is associated with a root mean square deviation (RMSD) of 7.7 Å (deactivated state) and 10.1 Å (ifenprodil-bound state). The movement of the α-C-atom of GluN2B F176 is characterized by a RMSD of 4.3 Å (deactivated state) and 7.5 Å (ifenprodil-bound state). The difference in RMSD values is caused by an 87° flip of the phenyl moiety from a vertical (active state) to a horizontal (deactivated/ifenprodil-bound state) position. This repositioning of the amino acid and its side chain together with the rearrangement of the complete GluN2B R2 subdomain are necessary for the formation of the ifenprodil binding site (Fig. 6b–d).

The crucial amino acid GluN2B F176 is at the beginning of a GluN2B R2 α-helix formed by residues GluN2B P177 to GluN2B N192. This helix is named α5 helix and connects the ATD with the LBD. Previous studies of different cryo-EM and also crystal structures showed the rearrangement of this α5 helix and the connected α5-α6 loop during the transition from an unbound and deactivated to the Ro 25-6981-bound state, which is caused by the movement of Glun2B R2[14,31]. In addition, the α5 helix of the homology models shows almost the same RMSD values of 4.7 Å (deactivated) and 7.4 Å (ifenprodil-bound) as the α-C-atom of GluN2B F176, suggesting a coupled movement during the transition. Whereas the applied transition forces in the morphing experiments from active to ifenprodil-bound state were smaller than from active to deactivated state, the resulting RMSD values for achieving the ifenprodil-bound state are higher than those necessary to reach the deactivated state. Reduced forces for a transition are consistent with favorable transition from active to the ifenprodil-bound state. The higher RMSD values are caused by a stronger downward movement of GluN2B F176 and α5 helix towards the LBD (Fig. 6b–e). These data suggest that ifenprodil and 3-benzazepines **1**–**7** bind to a hinge region of the ATD, which

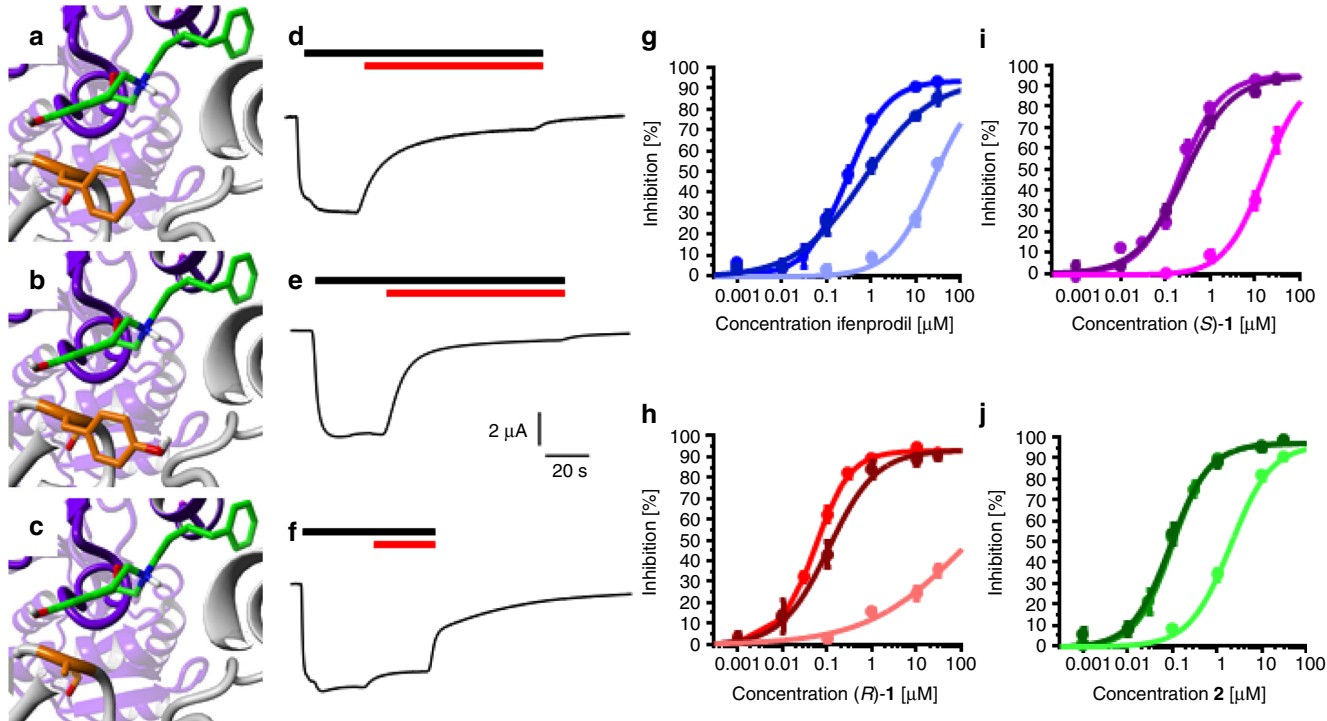

**Fig. 7** GluN2B wildtype vs. GluN2B F176A vs. GluN2B F176Y. Ifenprodil binding site of (**a**–**c**). GluN1-1a (purple)/GluN2B (gray) receptor with bound (*R*)-**1** and wildtype GluN2B F176 (**a**, orange), mutated GluN2B F176Y (**b**, orange) and mutated GluN2B F176A (**c**, orange). **d**–**f** Representative current traces of 1 μM (*R*)-**1** (red bar) inhibiting wildtype (**d**), GluN2B F176Y (**e**) and GluN2B F176A NMDARs in presence of 10 μM (*S*)-glutamate and 10 μM glycine (black bar). **g**–**j** Dose-response curves resulting from percental inhibition ± SEM of ifenprodil (**g**, blue), (*R*)-**1** (**h**, red), (*S*)-**1** (**i**, purple) and **2** (**j**, green) inhibiting GluN2B wildtype, GluN2B F176Y (dark color) and GluN2B F176A (light color) NMDARs. Curves were generated from recordings of *n* = 12–24 independent oocytes for each compound (see Supplementary Table 2–5)

undergoes extensive movements during the transition from active to deactivated state. In particular, the flip of GluN2B F176 side chain is prevented by these antagonists, subsequently preventing the α5 helix movement and the rearrangement of GluN2B R2 towards GluN1 ATD and GluN2B R1. Since rearrangements are only possible in the absence of ligands, it can be assumed that ifenprodil and 3-benzazepines **1**–**7** act like a "foot-in-the-door" by π/π-interactions with GluN2B F176.

**Confirmation of the "foot-in-the-door"-mechanism in vitro.** In order to confirm the "foot-in-the-door"-mechanism and the role of the crucial π/π-interactions between GluN2B F176 and the inhibitors, the phenylalanine was mutated to tyrosine (GluN2B F176Y). If the mechanism depends critically on the inhibition of side chain movement by aromatic interactions, the tyrosine mutant must recover the inhibitory activity of ifenprodil and the tested 3-benzazepines. Indeed, the results shown in Fig. 7 indicate similar effects for the wildtype NMDAR (GluN2B F176) and the tyrosine mutant. Therefore, it can be concluded that the π/π-interactions between the phenyl moiety of GluN2B F176 and the GluN2B inhibitors ifenprodil and 3-benzazepines (*R*)-**1**, (*S*)-**1** and **2** are the structural basis for the "foot-in-the-door" inhibition mechanism.

## Discussion

3-Benzazepines are promising derivatives for a new generation of ifenprodil-derived GluN2B-selective NAMs. The transformation of flexible into more rigid 3-benzazepine scaffold decreased the affinity at non-NMDARs without losing selectivity for GluN2B-containing NMDARs. However, an additional methyl group in 2-position increases the activity at GluN2C- and GluN2D-

containing NMDARs. 3-Benzazepine (*R*)-**1** represents a starting point for highly active and GluN2B-selective NAMs, since activity of this compound could be shown in vitro via TEVC and ex vivo via whole cell patch clamp in CA1-neurons in hippocampal slices.

In the competition assay, all tested 3-benzazepines **1**–**7** were able to displace the radioligand [³H]ifenprodil at nanomolar concentrations. This competition clearly indicates a common or at least overlapping binding site of ifenprodil and 3-benzazepines **1**–**7**. Docking studies demonstrated similar binding poses for ifenprodil and 3-benzazepines **1**–**7**. Moreover, the most active 3-benzazepines and ifenprodil adopt nearly identical orientations within the binding pocket like compounds from the 93-series, which are in accordance with the pharmacophore model despite a different scaffold[32,35].

Although 3-benzazepines **1**–**7** exhibited moderate to high affinity to the ifenprodil binding site, only (*R*)-**1** and **2** showed higher inhibitory activity at GluN1-1a/GluN2B receptors than ifenprodil. Moreover, affinity and activity do not correlate suggesting that only a specific binding mode with defined interactions evoke inhibitory activity. Based on activity data for (*R*)-**1**, (*S*)-**1**, and **2** it was postulated that a (*R*)-configured benzylic OH moiety is responsible for increased inhibitory activity compared to ifenprodil.

To uncover the mechanism, the binding site was dissected in five interactions zones, which were analyzed by mutagenesis of GluN1-1a and GluN2B subunits and variation of the substitution pattern of 3-benzazepines. The interaction hot spots resulting in high impact on compound activity are shown in Fig. 8. Hot spots of the binding pocket were characterized by high $IC_{50}$ shift factors of the corresponding mutants, while hot spots of the ligand substitution pattern were identified by strongly reduced activity of analogs compared to *rac*-**1**. In general, ifenprodil and 3-

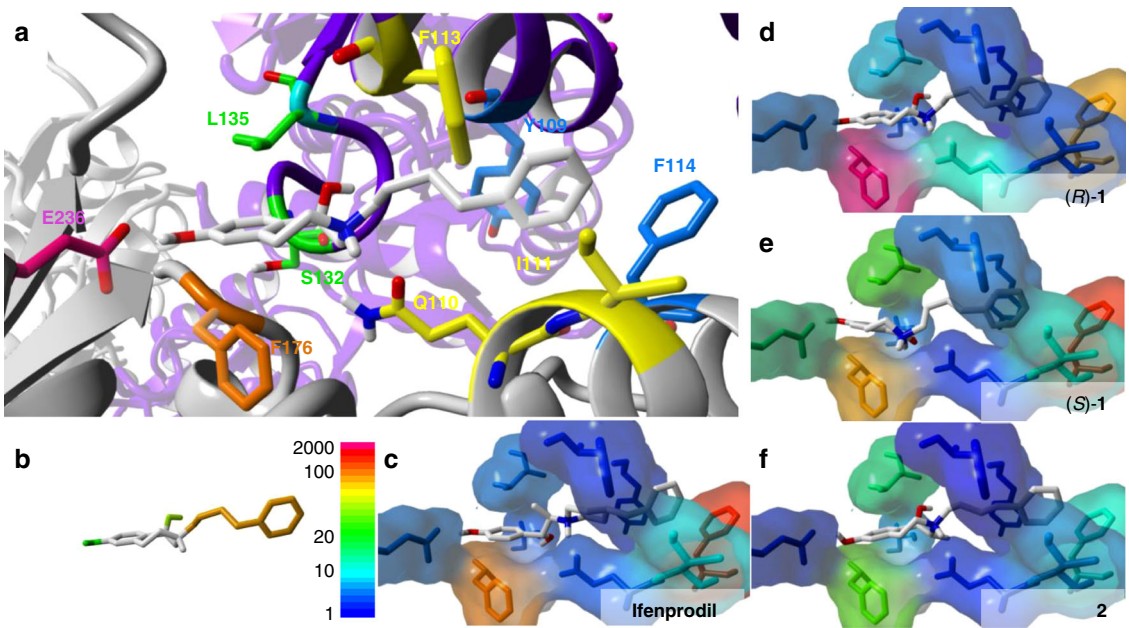

**Fig. 8 Interaction hot spots. a** Docking pose of (*R*)-**1** (CPK color with gray for C-atoms) at ifenprodil binding site of GluN1-1a (purple)/GluN2B (gray) with residues from IZ1 (magenta), IZ2 (green), IZ3 (blue), IZ4 (yellow) and IZ5 (orange). **b** Heat map for substitution pattern based on structure of (*R*)-**1** and scale bar for IC$_{50}$ shift factors displayed in Fig. 8c–f. Substituents are colored based on IC$_{50}$ shifts of **3**–**7** compared to *rac*-**1**. **c–f** Heat map for amino acid exchange based on the results of activity data for ifenprodil (**c**), (*R*)-**1** (**d**), (*S*)-**1** (**e**) and **2** (**f**) at mutated NMDARs compared to activity at wildtype NMDARs. Residue color based on IC$_{50}$ shift factors

benzazepines (*R*)-**1**, (*S*)-**1** and **2** show similar mutant dependency forming a triangle of hot spots between interaction zones IZ2, IZ3, and IZ5. This similar orientation of critical interaction zones indicates a common inhibition mechanism. The activity of highly potent compounds depends considerably on π/π-interactions with the aromatic amino acids GluN2B F114 and GluN2B F176, which is in clear accordance with the existing pharmacophore model[32,35]. Mutations of these amino acids to alanine led to dramatic loss of activity. Conformational restriction of the spacer between the amino moiety and the terminal phenyl ring resulted in very low activity of **6** and **7**. It was concluded that a particular flexibility of the side chain is required to orient properly in the binding pocket leading to simultaneous interactions with IZ1, IZ2, and IZ3. In addition to the proper orientation of the ligands in the binding pocket, the interaction of both OH moieties with IZ1 and IZ2 are required to achieve high inhibitory activity. Removal or modification of the benzylic (**3**), phenolic (**4**) or both OH moieties (**5**) led to considerably reduced activity.

Although mutations of the receptor reveal the same triangle of hot spots for ifenprodil, (*R*)-**1**, (*S*)-**1**, and **2**, differences can be seen in the order of impact for compound activity at the edges of the triangle (IZ2, IZ3, IZ5). While the highest IC$_{50}$ shift factors for ifenprodil and (*S*)-**1** were generated by the mutation of GluN2B F114, the highest shift factors for (*R*)-**1** and **2** were generated by the mutation of GluN2B F176 (Fig. 8c–f, Supplementary Tables 2–5). The differences in the relative dependencies on interactions with these two phenylalanines indicate small positioning differences of the ligands along the horizontal axis within the binding pocket modulating the strength of possible interactions. Additionally, ifenprodil and (*S*)-**1** also behave more similar against the other mutants than (*R*)-**1** and **2**, which could explain their similar activity. Moreover, these results indicate that increased interactions with GluN2B F176 compared to GluN2B F114 could be the reason for higher activities of (*R*)-**1** and **2**.

The activity of **2** is less dependent on interactions with the GluN2B subunit than the activity of (*R*)-**1**, (*S*)-**1** and ifenprodil. **2** shows an increased influence of IZ2 and reduced influence of the

GluN2B interaction zones IZ3 and IZ5. This could be caused by additional hydrophobic interactions of the methyl moiety in 2-position with the isobutyl residue of GluN1-1a L135. Reduced GluN2B interactions can explain loss of selectivity towards GluN2C- and Glun2D-containing NMDARs. In summary, all four compounds reveal similar critical interactions, but small differences in the order of influence of these interactions exist.

It is concluded that the high activity of ifenprodil and 3-benzazepines (*R*)-**1**, (*S*)-**1** and **2** is based on aromatic interactions with both phenyl moieties of GluN2B F114 and F176 and polar interactions of the benzylic and phenolic OH moieties with the interaction zones IZ1 and IZ2. Although both OH groups of 3-benzazepines and ifenprodil contribute to their high activity, the π/π-interactions are more crucial for the mechanism of inhibition.

Our in silico experiments led to a hypothesis of mechanism, that is in good accordance with previous and current in vitro data and the existing pharmacophore model[32,35]. The binding pocket is located within a hinge region in the ATD of the NMDAR. Since the binding pocket is only formed in the deactivated state, ifenprodil and related 3-benzazepines bind to this conformation, stabilize it and inhibit the transition into the active state (Fig. 9). A similar state-dependent inhibition mechanism was previously postulated for ifenprodil[30].

Docking and mutagenesis studies led to the conclusion that the state-dependent binding of ifenprodil and 3-benzazepines is driven by the interactions with IZ1–IZ3 requiring a defined ligand conformation. On the other hand, inhibition of the state transition is based on the π/π-interactions between the ligand and the phenyl moiety of GluN2B F176 in IZ5. Simultaneous interactions via 3-benzazepines or ifenprodil with IZ1, IZ2, IZ3, and IZ5 restrain a structural connection among GluN1 ATD and the two adjacent GluN2B ATD subdomains.

Simulation of the inhibition process reveal distinct movements of the entire GluN2B F176 residue and repositioning of its phenyl ring from a vertical to a horizontal position. This movement is directly correlated with the downward movement of the α5 helix

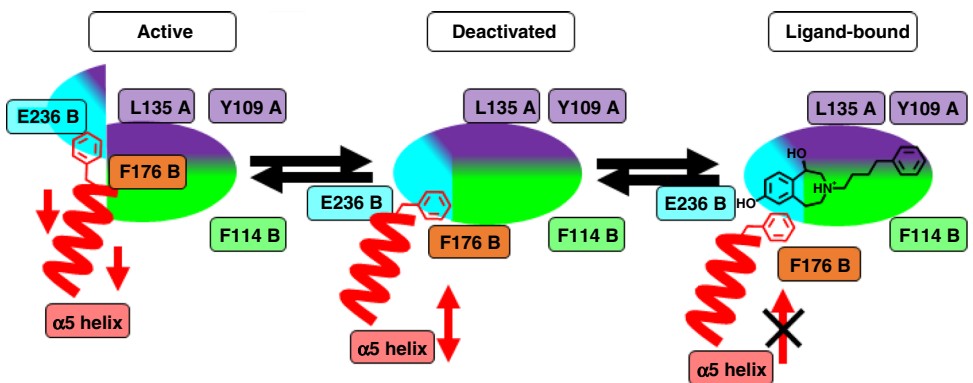

**Fig. 9** Mechanism of inhibition for ifenprodil and 3-benzazepines. Mechanism of inhibition for ifenprodil and related 3-benzazepines at the ifenprodil binding site built by amino acids from the GluN1 ATD (purple) and the GluN2B R1 (green) and GluN2B R2 (cyan) subdomain. In active conformation phenyl moiety of GluN2B F176 (red) is orientated vertically and the GluN2B R2 subdomain is displaced preventing ligand binding. Transition from active to deactivated state results in repositioning of GluN2B R2 subdomain, moving α5 helix (red) downwards and subsequently reorient GluN2B F176 phenyl moiety to horizontal position enabling ligand binding. After compound binding to the receptor, backward movement of GluN2B F176 and α5 helix is inhibited resulting in structural coupling of GluN2B ATD subdomains with the GluN1 ATD, which prevents opposing movements resulting in locked receptor conformation

during the transition from the active to the deactivated state, which is enabled by the independent movement of GluN2B R2 towards GluN1 and GluN2B R1. As a result of these concerted movements, the backward movement of the α5 helix is prevented by the π/π-interactions between the ligands and the phenyl ring of GluN2B F176 locking the receptor in the deactivated state (Fig. 9).

The downward movement of α5 helix and the movement of GluN2B ATD subdomains, was also observed in cryo-EM studies of unbound and Ro 25-6981-bound receptor conformations[31]. Moreover, the importance of α5 helix for receptor transition and interdomain interactions was also shown in previous crystal structure studies[14]. Since ifenprodil and Ro 25-6981 are structurally very similar and interact with the same interaction partners inside the binding pocket, a common mechanism of inhibition is postulated for both ligands.

The importance of the π/π-interactions of GluN2B F176 and the ligands was shown by mutation of phenylalanine into tyrosine and recording the inhibitory activity of the ligands at this mutated receptor. The same behavior of the ligands at this GluN2B F176Y mutant confirms the role of the π/π-interactions with this amino acid.

Furthermore, this mechanism of inhibition is well supported by published crystal structures of seven different ligands from the 93-series (PDB 6E7R-6E7X) showing all π/π-interactions with GluN2B F176 with distances between ligand and phenyl ring less than 5.0 Å[36].

In summary, it is postulated that all compounds enter the ifenprodil binding site during the deactivated state. The ligands adopt an almost linear pose in the binding site forming two H-bonds and two π/π-stacking interactions in four interaction zones IZ1–IZ3 and IZ5. While simultaneous interactions with IZ1–IZ3 are responsible for fixation of the ligand in the binding pocket, the interaction with IZ5 is the basis of the "foot-in-the-door"-mechanism: After ligand binding the backward movement of GluN2B R2 subdomain is blocked by the aromatic interactions. This "foot-in-the-door"-mechanism locks the receptor in the deactivated state by inhibition of GluN2B F176 and α5 helix backward movements until the inhibitor is released again.

## Methods

**Synthesis, compound solutions, and TEVC-recordings**. The compounds **1**, **2**, and **4–7** were previously reported[33,34,37–39]. Synthesis of compound **3** is described in the Supplementary Information (Supplementary Fig. 2 and Supplementary

Methods). Ifenprodil (+)-tartrate was commercially available by Sigma Aldrich. Test compounds were used as 10 mM solutions in DMSO, which were diluted with agonist solution to obtain the desired concentration of compound and constant concentration of agonists. All test solutions were adjusted to DMSO concentrations of 0.1%. Compound activity was tested by two-electrode voltage clamp (TEVC) in *Xenopus laevis* oocytes at room temperature as described before[34]. For measurements, oocytes were constantly superfused with $Ba^{2+}$ Ringer (pH 7.4) containing (mmol $L^{-1}$): 10 HEPES, 90 NaCl, 1 KCl, 1.5 $BaCl_2$. Agonist solution containing 10 μM glycine and 10 μM (S)-glutamate was prepared from 100 mM stock solutions in $Ba^{2+}$ Ringer for each measurement. Measurements were performed at a holding potential of −70 mV and recording pipettes (0.5–1.5 MΩ) were backfilled with 3 M KCl. Each compound concentration was tested at three different oocytes resulting in 12–24 oocytes for every dose-response curve depending on tested concentration levels

**Molecular biology and oocyte preparation**. cRNAs, DNA constructs and oocytes were prepared as previously described[34]. GluN1-1a and GluN2B mutants were generated and verified by BioCat GmbH (Heidelberg, Germany) or Thermo Fisher Scientific GENEART GmbH (Regensburg, Germany). Defolliculated oocytes were obtained from EcoCyte Bioscience (Dortmund, Germany). Each oocyte was injected with 0.8 ng GluN1 (wildtype or mutant) and 0.8 ng GluN2B subunit (wildtype or mutant) cRNA using a nanoliter injector 2000 (WPI, Berlin, Germany). For activity recordings at NMDARs with different GluN2 subunits oocytes were injected with 0.8 ng GluN1-1a and 0.8 GluN2A cRNA or 2 ng GluN1-1a and 4 ng GluN2C or GluN2D cRNA. cRNAs were generated by in vitro transcription with the Ambion T7 mMessage mMachine kit (Life Technologies, Darmstadt, Germany) from linearized cDNA templates (wildtype rat, linearized with restriction enzymes PacI for GluN1/pSGEM, GluN2A/pSGEM and GluN2C/pSGEM or NheI for GluN2B/pSGEM and GluN2D/pSGEM). Injected oocytes were stored for 4–6 d at 18 °C in Bath's solution containing (mmol $L^{-1}$): 88 NaCl, 1 KCl, 0.4 $CaCl_2$, 0.33 $Ca(NO_3)_2$, 0.6 $MgSO_4$, 5 TRIS-HCl, 2.4 $NaHCO_3$ and supplemented with 80 mg $L^{-1}$ theophylline, 63 mg $L^{-1}$ benzylpenicillin, 40 mg $L^{-1}$ streptomycin, and 100 mg $L^{-1}$ gentamicin.

**In vitro slice preparation and voltage clamp recordings**. All animal work has been approved by local authorities (review board institution: Landesamt für Natur, Umwelt und Verbraucherschutz Nordrhein-Westfalen; approval ID: 84-02.05.50.15.026). Experiments were performed on C57BL/6J mice ranging in age from postnatal day P30 to P40. After sacrificing the mice, a block of brain tissue containing the hippocampus was removed from the cranial vault and submerged in ice-cold aerated ($O_2$) saline containing (in mM): sucrose, 200; PIPES, 20; KCl, 2.5; $NaH_2PO_4$, 1.25; $MgSO_4$, 10; $CaCl_2$, 0.5; dextrose, 10; pH 7.35, with NaOH. In total, 300 μm thick horizontal hippocampal slices were prepared on a vibratome. Slices were transferred to a holding chamber and kept submerged (at 30 °C for 30 min, thereafter at room temperature) in artificial cerebrospinal fluid (ACSF) containing (in mM): NaCl, 125; KCl, 2.5; $NaH_2PO_4$, 1.25; $NaHCO_3$, 24; $MgSO_4$, 2; $CaCl_2$, 2; dextrose, 10; pH adjusted to 7.35 by bubbling with carbogen (95% $O_2$ and 5% $CO_2$ gas mixture).

Whole-cell patch clamp recordings were performed on CA1 neurons using an EPC-9 amplifier (HEKA Elektronik, Lamprecht, Germany). Voltage-clamp experiments were controlled by the PatchMaster software (HEKA Elektronik). Electrodes had resistance of 2.5–3.5 MΩ. Access resistances were between 8 and 20

MΩ. Recordings were done in ACSF containing (in mM): NaCl, 125; KCl, 2.5; NaH$_2$PO$_4$, 1.25; NaHCO$_3$, 24; MgSO$_4$, 2; CaCl$_2$, 2; dextrose, 10; pH adjusted to 7.35 by bubbling with carbogen. The whole-cell pipette solution contained (in mM): NaCl, 10; CsMeSO$_4$, 110; EGTA, 11; HEPES, 10; KCl, 1; TEA-Cl, 10; QX314-Cl, 3.35; MgCl$_2$, 1; CaCl$_2$, 1; Mg-ATP, 3; Na-GTP, 0.5. pH of the internal solution was set to 7.25 with CsOH and an osmolality was 295–300 mOsm kg$^{-1}$. Evoked EPSCs were recorded following stimulation of Schaffer collaterals using a bipolar microelectrode. The stimulation consisted of 100 μs current pulses (0.1–1 mA) in order to evoke currents of about 150 pA and were repeated every 30 s. Recordings were done at 30–32 °C. The NMDA receptor-dependent component of the EPSC was pharmacologically isolated. GABAergic transmission was blocked by 25 μM gabazine (GABA$_A$R antagonist) and 10 μM CGP 55845 (GABA$_B$R antagonist). When EPSCs were recorded using a CsMeSO$_4$ containing intracellular solution, outwards currents at +40 mV were mediated by a fast AMPA receptor-dependent and a slower NMDA receptor-dependent component. In order to block the fast component, 10 μM NBQX (AMPA receptor antagonist) was added to the recording solution, spearing a slow outward current reminiscent of a NMDA receptor-mediated current.

### Data analysis

Electrophysiological data were recorded and analyzed with GePulse and Ana (Dr. Michael Pusch, Genova, Italy; http://users.ge.ibf.cnr.it/pusch/). Data analysis was done using OriginPro 2018 (OriginLab Corporation, Northampton, MA, USA). Inhibitory activity at a defined compound concentration was calculated as previously described by the following equation:[34]

$$\text{inhibition} = 1 - \frac{I_c - I_h}{I_a - I_h}$$

$I_h$ represents the holding current of oocyte without agonists; $I_a$ is defined as the steady-state current after activation by 10 μM glycine and 10 μM (S)-glutamate; $I_c$ is the resulting steady-state current in presence of agonists and compound. Dose-response curves were fitted to the logistic sigmoid equation:

$$y = \frac{A1 - A2}{1 + \left(\frac{x}{x_0}\right)^P} + A2$$

A1 and A2 are defined as the minimal and maximal inhibition by a compound. While A1 was determined as A1 = 0%, A2 was kept flexible. If necessary, A2 was fixed to the A2 value of GluN1-1a/GluN2B wildtype inhibition; $p$ is defined as the slope of the curve; $x_0$ is the concentration at half-maximal inhibition and $x$ is the tested concentration. For mutant analysis the IC$_{50}$ shift factor was calculated by the following equation:

$$\text{IC}_{50}\text{ shift factor} = \frac{\text{IC}_{50}[\text{mutated receptor}]}{\text{IC}_{50}[\text{wildtype receptor}]}$$

### Cell culture and receptor binding studies

Receptor binding studies were performed as previously described[40]. In brief dexamethasone-inducible GluN1-1a~/GluN2B expressing mouse L(tk) cells were used for preparation of the membrane fraction. For induction and subsequent binding assay, the original growth medium was replaced by growth medium containing 4 μM dexamethasone and 4 μM ketamine (final concentration). After 24 h, the cells were rinsed with phosphate buffered saline solution (PBS, Biochrom AG, Berlin, Germany), harvested by mechanical detachment and pelleted. For binding assay, cell pellet was resuspended in PBS solution and number of cells was determined using a Scepter® cell counter (MERCK Millipore, Darmstadt, Germany). Cell fragments were generated by sonication and centrifugation (23,500 × g, 4 °C) resulting in fragments of ~500,000 cell/mL (23,500 × g, 4 °C). The suspension of membrane homogenates was sonicated again (4 °C, 2 × 10 s cycles with a break of 10 s) and stored at −80 °C. The competitive binding assay was performed with the radioligand [3H]ifenprodil (60 Ci mmol$^{-1}$; BIOTREND, Cologne, Germany). The thawed cell membrane preparation (about 20 μg protein) was incubated with various concentrations of test compounds, 5 nM [3H]ifenprodil, and TRIS/EDTA-buffer (5 mM TRIS/1 mM EDTA, pH 7.5) at 37 °C. The non-specific binding was determined with 10 μM unlabeled ifenprodil. The Kd value of ifenprodil is 7.6 nM.

### Docking, homology modeling, and morphing

Docking experiments, homology modeling, and morphing were performed using YASARA 18, implemented and adapted macros, and previously published cryo-EM and crystal structures with PDB numbers 4PE5, 5FXG, and 5FXI[22,41–43]. For docking, ligands were constructed in YASARA 18 with defined stereochemistry leading to a library containing 13 different ligands (see Supplementary Information). Docking was performed using crystal structure 4PE5 with a simulation box surrounding the binding site and modified YASARA macro "dock_runscreening.mcr" with 50 runs per ligand, AMBER14 force field and AutoDockLGA method. For subsequent analysis of interacting residues, the docking conformation with highest binding energy was chosen for each ligand and compared to the binding conformation of ifenprodil in the crystal structure (PDB 4PE5, Supplementary Table 6). Residues identified as crucial for ligand binding were selected for mutagenesis. Homology modeling was performed using YASARA 18 macro "hm_build.mcr" using PDB structures 4PE5, 5FXG, and 5FXI[44,45]. Sequences for homology models were taken from GenBank entries U08263.1 (GluN1-1b, rat) and U11419.1 (GluN2B, rat). For morphing experiments, homology model alignments for all three states with highest Z-Score were selected. Morphing procedure is defined as a low-energy pathway between the initial and final conformation. The low-energy transition was calculated using YASARA 18, implemented force field AMBER03 and chosen homology models of active (PDB 5FXG), deactivated (PDB 5FXI) and ifenprodil-bound (PDB 4PE5) states. The morphing procedure started with removing clashes with an unrestrained steepest descent minimization followed by a restrained annealing minimization. Morphing forces were applied to all atoms starting from 10 to 20,000 pN until convergence of the active and deactivated/ifenprodil-bound model was reached. The steered simulation was calculated with time steps of 2 fs and progress was shown in 150 snapshots. Every 10th step, atom velocities were scaled down by 0.9 and RMSD of target conformation was measured. If RMSD changed less than 0.01 Å, the pulling force was increased by 50%. To avoid simulation failures caused by too strong pulling, simulation results were analyzed at snapshots 0–120 where pulling forces lower than 2000 pN were applied.

### Statistics and reproducibility

All oocyte measurements were performed at least at three independent oocytes for every compound and every concentration leading to a number of $n = 3$–24 depending on the number of tested concentrations for every compound. Exact values for all experiments are given in Table 1 and supplementary Tables 1, 3, 4, 5 and 6. For affinity measurements all data were collected from three independent experiments at different days. Whole cell patch clamp data were derived from $n = 5$ independent experiments.

If applicable, significance of data was tested by one-way ANOVA and Tukey post-hoc mean comparison test and indicated by ***$p$-values < 0.001, **$p$-values < 0.01, *$p$-values < 0.05 and ns for $p$-values > 0.05.

**Reporting summary**. Further information on research design is available in the Nature Research Reporting Summary linked to this article.

## Data availability

Source data are available in Figshare (https://doi.org/10.6084/m9.figshare.9751739.v1; https://doi.org/10.6084/m9.figshare.9751751.v1; https://doi.org/10.6084/m9.figshare.9751742.v1; https://doi.org/10.6084/m9.figshare.9751745.v1; https://doi.org/10.6084/m9.figshare.9751748.v1; https://doi.org/10.6084/m9.figshare.9751736.v1)[46–51] All other data supporting the finding of this study are included in the article and the supplementary information.

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

## Acknowledgements

This work was supported by Cells-in-Motion (CiM) Cluster of Excellence, in particular by the pilot project PP-2016-23 - The role of the NMDA receptor subunit 2B (GluN2B) in the regulation of autoimmune neuroinflammation and characterization of novel selective antagonists. Financial support by the Deutsche Forschungsgemeinschaft is gratefully acknowledged. We also thank Elmar Krieger for the implementation of force analysis of morphing to the program YASARA.

## Author contributions

J.A.S. designed and performed TEVC measurements, docking experiments, homology modeling and morphing. D.S. and B.F. conceived and performed affinity and binding studies. S.T. synthesized compound **3**. M.D. and T.B. performed and analyzed EPSC measurements in CA1 neurons. M.H. generated DNA wildtype constructs for TEVC-measurements. N.S.S. helped with mutant design and molecular biology procedures. G.S. and B.W. are the supervisors of this work. The manuscript was written by J.A.S., G.S. and B.W. with the help of M.D., T.B., N.S.S. and M.H.

## Competing interests

The authors declare no competing interests.
