## [Peer Review File · Communications Biology]

Reviewers' comments:

Reviewer #1 (Remarks to the Author):

In the manuscript prepared by Schreiber et al., the authors divided the binding pocket of GluN2B-selective inhibitors into distinct interaction zones, and then carefully examined the roles of residues in these interaction zones. With mutagenesis and different inhibitors, the authors revealed residues responsible for inhibitor binding and residues responsible for inhibitor efficacy. These results suggested a foot-in-the-door mechanism for inhibition of GluN1/2B by ifenprodil and analogs in which binding of antagonist prevents the transition of conformations from inactive state to active state of the receptor. It is an interesting study with good hypotheses and substantiated conclusions. Some concerns should be addressed to improve the manuscript.

1. Please show dose-response traces of TEVC recordings in figure 1b instead. Numbers of recorded oocytes should be given in figures or figure legends. Please state the reason for choosing the non-saturating concentration of glutamate (10 μ M) to activate GluN1/2B receptors.
2. Do 3-benzazepines 1 and 2 show selective inhibition at GluN1/2B over GluN1/2A, GluN1/2C, and GluN1/2D?
3. F114 and E236 in GluN2B are conserved among GluN2A-2D. F176 is conserved in GluN2A and GluN2B. Please speculate on why ifenprodil and analogs are selective for GluN1/2B over GluN1/2A, GluN1/2C, and GluN1/2D receptors?
4. The authors should elaborate on how their results compare to the findings in two previous papers: Burger et al. *Mol Pharmacol* 82(2): 344-359 (2012) and Yuan et al. *Neuron* 85(6): 1305-1318 (2015).

Reviewer #2 (Remarks to the Author):

The manuscript from Schreiber and colleague describe in an elegant series of both in vitro and in silico experiments a novel inhibitory mechanism of ifenprodil and the related compound 3-benzazepines.

The overall manuscript is sound and novel; it is in general of high interest because of the potential therapeutic uses of ifenprodil and its related compounds.

Few observations:

The inhibitory mechanism of the 3-benzazepines should be tested in vitro also towards GluN1-GluN2A receptors to verify the specificity of the inhibitory mechanism/s

More in general although the study is elegant a validation of the inhibitory activity of the novel compounds in more physiological context would have been important to test. Did the authors perform some experiments of IC50 at least in neuronal cell cultures? These experiments would increase the interest of the manuscript

Reviewer #1 (Remarks to the Author):

In the manuscript prepared by Schreiber et al., the authors divided the binding pocket of GluN2B-selective inhibitors into distinct interaction zones, and then carefully examined the roles of residues in these interaction zones. With mutagenesis and different inhibitors, the authors revealed residues responsible for inhibitor binding and residues responsible for inhibitor efficacy. These results suggested a foot-in-the-door mechanism for inhibition of GluN1/2B by ifenprodil and analogues in which binding of antagonist prevents the transition of conformations from inactive state to active state of the receptor. It is an interesting study with good hypotheses and substantiated conclusions. Some concerns should be addressed to improve the manuscript.

Please show dose-response traces of TEVC recordings in figure 1b instead. Numbers of recorded oocytes should be given in figures or figure legends.

Response: Several GluN/pSGEM-constructs showed run-up-, run-down-phenomena and desensitization of activated NMDA receptors in oocytes. These phenomena have a negative influence on the reproducibility and accuracy of inhibition measurements, especially at long periods of receptor activation. Due to these phenomena dose response curves in one oocyte could not be conducted. Instead, we decided to invest a multitude of work and measure just one concentration of one compound at one oocyte to reduce the influence of these phenomena and generate data with higher accuracy. Using this approach each individual data point is based on inhibition data of at least 3 independent oocytes. Complete dose-response-curves were generated from data of 12 - 24 independent oocytes. To clarify that we used this approach, the number of used oocytes is now added to Figure legends and Tables.

Please state the reason for choosing the non-saturating concentration of glutamate (10 μ M) to activate GluN1/2B receptors.

Response: Ifenprodil and related compounds show only weak agonist dependency. 10 μ M of (S)-glutamate led to almost 100 % response.¹ As a compromise, to avoid

long activation and still allow for fast washouts of the agonists, the amount of glycine and (S)-glutamate was reduced to 10 μ M.

Do 3-benzazepines 1 and 2 show selective inhibition at GluN1/2B over GluN1/2A, GluN1/2C, and GluN1/2D?

Response: Inhibitory activity of 10 μ M (*R*)-1, (*S*)-1 and 2 were determined at GluN1/2A, GluN1/2C and GluN1/2D receptors by TEVC at 3 (GluN2B) or 5 (GluN2A, GluN2C, GluN2D) independent oocytes and were compared to the inhibitory activity of 10 μ M ifenprodil. Results were added to the manuscript and the supporting information. The enantiomers of 1 did not show significant differences in selectivity over the different GluN2 subunits compared to ifenprodil. However, the additional methyl group in 2-position of compound 2 led to reduced selectivity over GluN2C and GluN2D. We speculate that the loss of activity is generated by a repositioning of 2 inside the binding site resulting in stronger interactions with GluN1 and weaker interactions with GluN2B subunit. This is in accordance with the results of the IC₅₀ factors for 2 (see heat maps in figure 8). Nevertheless 10 μ M of compound 2 led to ion current inhibition of 56 ± 2 % (SEM, GluN1/2C) and 55 ± 2 % (SEM, GluN1/2D) compared to an IC₅₀ of 91 ± 7 nM at GluN1-1a/GluN2B NMDARs.

F114 and E236 in GluN2B are conserved among GluN2A-2D. F176 is conserved in GluN2A and GluN2B. Please speculate on why ifenprodil and analogues are selective for GluN1/2B over GluN1/2A, GluN1/2C, and GluN1/2D receptors?

Response: The activity of ifenprodil and analogues depends on 1) the possibility of binding to a suitable sized binding pocket and 2) the ability to block α 5 helix movement via the interaction with the phenylalanine 176. In case of GluN2D-containing NMDARs the most critical aromatic interaction between the ligands and the GluN2-subunit does not exist, since GluN2D-subunit bears an alanine instead of phenylalanine. The GluN2C-subunit has a histidine at the position of GluN2B F176. Although histidine bears an aromatic imidazole ring, the chemical properties of this amino acid are different to phenylalanine and may not completely substitute for the phenylalanine aromatic interactions. Although GluN2A subunit bears a phenylalanine as well, the geometry of the predicted binding site is considerably different as estimated from structure 5FXI. Specifically, geometry and volume are not

appropriate to allow binding of the ligands in the proposed binding pose. The reduction of binding site volume can be seen as well in the Cryo-EM structure 5UOW showing a trimeric GluN1/GluN2A/GluN2B NMDAR. Nevertheless, these hypotheses are highly speculative and they would need more structural data for all the different GluN2 subunit-containing NMDARs. Based on the available data we feel not confident to include a hypothesis for selectivity of these compounds into the manuscript.

The authors should elaborate on how their results compare to the findings in two previous papers: Burger et al. *Mol Pharmacol* 82(2): 344-359 (2012) and Yuan et al. *Neuron* 85(6): 1305-1318 (2015).

Response: Results are now compared to the two mentioned studies. Explicitly we have discussed these papers at important points of the manuscript in relation to the pharmacophore model, the interaction between protonated amine and GluN2B Q110 and the aromatic interactions with GluN2B F114.

Reviewer #2 (Remarks to the Author):

The manuscript from Schreiber and colleague describe in an elegant series of both in vitro and in silico experiments a novel inhibitory mechanism of ifenprodil and the related compound 3-benzazepines.

The overall manuscript is sound and novel; it is in general of high interest because of the potential therapeutic uses of ifenprodil and its related compounds.

Few observations:

The inhibitory mechanism of the 3-benzazepines should be tested in vitro also towards GluN1-GluN2A receptors to verify the specificity of the inhibitory mechanism/s

Response: See response to reviewer 1. Selectivity towards GluN2A, GluN2C and GluN2D subunit-containing NMDARs was examined for (*R*)-1, (*S*)-1 and 2 by TEVC-experiments and compared to selectivity of ifenprodil.

More in general although the study is elegant a validation of the inhibitory activity of the novel compounds in more physiological context would have been important to test. Did the authors perform some experiments of IC50 at least in neuronal cell cultures? These experiments would increase the interest of the manuscript.

Response: In order to address this point, we performed whole-cell patch clamp recordings on CA1 neurons in murine hippocampal slices with our most active compound (*R*)-1 and ifenprodil for comparison. Application of both compounds at CA1 neurons led to significant reduction of EPSC signal. Indeed, these are the first ex vivo data for a 3-benzazepine-based compound. Of note, an animal study (rat) on activity of 3-benzazepine-based compounds is currently conducted by our cooperation partners.

Reviewer #3

A common foot-in-the-door mechanism allows selective targeting of N-methyl-D-aspartate receptors

By: Schreiber et al.

In this paper the authors use the ifenprodil scaffold to obtain seven 3-benzazepine compounds which inhibit diheteromeric GluN2B NMDA receptors. Using these new compounds, the authors investigate the binding site on the GluN2B subunit N-terminal domain and postulate a mechanism for the antagonist binding. Overall this is an interesting paper. However, there are a number of aspects that I feel need to be revised.

Specific comments:

Title: I'm sorry to say I think the 'foot-in-the-door' phrase is not a good terminology for this mechanism. This term has been associated with the ion channel blocking mechanism of drug action for the past 30-40 years. Therefore, I suggest the authors should not use this mis-leading terminology in their title.

Response: The title of the manuscript has been revised.

Abstract: The abstract states, 'Structural reorganization of the ifenprodil scaffold to obtain 3-benzazepines retained inhibitory GluN2B activity but increased the selectivity towards these non-GluN2B NMDARs.'. This phrasing is confusing. Why say increased selectivity towards these non-GluN2B NMDARs? Which non-GluN2B NMDARs are being considered here. Surely it would be clearer to say 'selectivity for GluN2B NMDARs'? However, selectivity evidence seems to be lacking in the paper. What is the IC50 of the new compounds for 2A, 2C or 2D receptors? What about inhibition of AMPA or Kainate receptors? Ifenprodil has a problem of 5-HT receptor inhibition. Were the new compounds tested at 5-HT receptors? If no selectivity data is given in the paper, no conclusions can be made about selectivity.

Response: The selectivity profile of 3-benzazepines was evaluated in previous studies by affinity measurements at non-NMDA-receptors, in particular at σ_1 and σ_2 receptors. These studies are reported in references of the manuscript²⁻⁵. Moreover, the interaction of *rac-1* with more than 100 relevant receptors, transporters and enzymes including adrenergic (α_1), dopaminergic (D1, D2), opioid (κ , μ , δ) and serotonergic receptors (5-HT_{1A}, 5-HT₂) was investigated by competition assays with suitable radioligands.² At a concentration of 1 - 3 μ M *rac-1* did not compete significantly with the employed radioligands proving the low off-target binding and the high selectivity for GluN2B-containing NMDARs. For NMDARs bearing different GluN2 subunits the activity of (*R*)-**1**, (*S*)-**1** and **2** was evaluated by TEVC measurements. The activity was compared to the activity of ifenprodil and the results have been added to the manuscript (see Supplementary Information). Both enantiomers of **1** did not show significant activity differences at GluN2A, GluN2C or GluN2D subunit-containing NMDARs compared to the activity of ifenprodil at these NMDARs. However, the additional methyl group in 2-position (**2**) slightly increases the activity at GluN2C and GluN2D subunit-containing NMDARs. At a concentration of 10 μ M **2** generates ion current inhibition of 56 ± 2 % (SEM; GluN2C) and 55 ± 2 % (SEM; GluN2D), which indicates at least 100-fold lower activity than at GluN2B subunit-containing NMDARs (IC_{50} 91 ± 7 nM).

Introduction, page-4, line 17: here is stated, 'ifenprodil reveals high selectivity towards NMDARs with other GluN2 subunits'. This is confusing. Ifenprodil is fairly selective for GluN2B receptors. It does not have selectivity towards other GluN2 subunits. Which other subunits are the authors talking about?

Response: The sentence has been rephrased to clarify our intention. The selectivity profile of ifenprodil is quite good. Nevertheless, the usage as a neuroprotective agent is limited because of the activity at some non-NMDARs (see comment below).

At end of the same paragraph on page-4 is stated, 'One reason for low selectivity towards non-NMDARs could be the high flexibility of the ifenprodil scaffold allowing adaptation to other binding sites as well.' This is an odd

thing to say, as ifenprodil selectivity for GluN2B, compared to other NMDA and non-NMDA receptors is quite good (Traynelis et al., 2010; Table 13).

Response: We mainly agree with this statement and revised the sentence to clarify our intention. Even if selectivity of ifenprodil for GluN2B subunit-containing NMDARs is quite good, ifenprodil shows activity at serotonergic and adrenergic receptors, which is the reason why ifenprodil was used as a vasodilator (Dilvax[®], Vadilex[®]). This activity at the mentioned receptors would lead to unwanted vascular side effects if ifenprodil is used as a neuroprotective therapeutic agent. In addition, ifenprodil shows high σ_1 and σ_2 receptor affinity. Therefore, it is needed to reduce the adrenergic activity and to optimize the selectivity profile of ifenprodil.

Introduction, page-5, line 6: 'To investigate the molecular mechanism and improve the selectivity towards non-glutamate receptors' This is confusing: why do the authors wish to improve selectivity towards non-glutamate receptors? Surely the aim should be to improve selectivity for GluN2B receptors and reduce off-target binding?

Response: We thank the reviewer for the comment and revised the sentence. The aim of structure optimization is to reduce the activity at non-NMDARs, especially adrenergic, serotonergic and sigma receptors.

Figure-9, Mechanism of Inhibition. This illustration is conceptually ok, but seems to ignore the relevance of proton inhibition?

Response: The illustration is a simplification of our results for the newly found inhibition mechanism of ifenprodil and 3-benzazepines. All TEVC measurements were performed at the same pH of 7.4. Therefore, the influence of different proton concentrations on the activity of the 3-benzazepines was not investigated. While the inhibition of the NMDA receptor by H⁺ and the influence of H⁺ concentration on ifenprodil inhibition is known for long time, the molecular basis and the structural interactions for these phenomena are still not completely understood. Different studies showed that the ATD of NMDARs undergoes distinct structural reorientation when the pH was reduced.⁶ The structural changes within the ATD also influence the

structure of the binding pocket, which could result in reduced binding probability for the ifenprodil-derived compounds. Nevertheless, these speculations would need further experiments to find the missing link between proton inhibition and activity of ifenprodil-derived compounds. Therefore, we decided to reduce the illustration to the essential findings of this study.

Table 1: Ifenprodil is an allosteric modulator whose binding triggers an increase in agonist affinity. Le Chatelier's principle therefore suggests that in the absence of agonist, the measured ifenprodil K_i will be of lower affinity than the IC_{50} measured in a functional assay. Therefore, the marked differences in K_i and measured IC_{50} values for some compounds is a surprise. What is the reason for this?

Response: We agree with the reviewer that the relationship between the affinity and the activity of the 3-benzazepines is curious. However, there are two major reasons which can explain these unexpected results. First, the affinity is measured indirectly by the ability of the 3-benzazepines to displace the radioligand $^3[H]$ ifenprodil while the activity is measured directly by the ability to inhibit the NMDA receptor. For the displacement of the radioligand, the 3-benzazepines need to bind in a similar conformation at an at least overlapping binding site. Also, the binding of the 3-benzazepine must be energetically more favourable than the binding of the radioligand. Our docking results indicate that the 3-benzazepines can adopt different orientations within the binding site, which are not favourable to displace the radioligand. Second, our results clearly show that the inhibitory activity is not generated by the binding itself. The activity is mainly generated by the blockade of the reorientation of the amino acid GluN2B F176 and the $\alpha 5$ helix during the receptor transition. To prevent this reorientation, simultaneous and strong interactions with IZ1-IZ3 are needed. If one interaction with these zones is missing, the binding energy of the compound is not strong enough to prevent the movement. This would cause the displacement of the 3-benzazepine from the binding site when the transition and the movements take place and thus a strong reduction in activity of the compound. Since these defined interactions and the ability to prevent the GluN2B

F176 movement are not requirements for the binding in the deactivated conformation, the affinity can be uncoupled from the activity.

The structural modelling could also refer to Lee et al (2014). NMDA receptor structures reveal subunit arrangement and pore architecture. Nature 511,191–197.

Response: The mentioned publication by Lee et al is now cited in the manuscript.

Page 17: section on, 'Inhibitory mechanism was uncovered and confirmed.....'. This section is confusing, because it refers to 'deactivated state' and 'active conformation'. It is not clear whether the authors are suggesting 'active conformation' means 'open channel'? Or does active conformation mean the active conformation of the N-terminal domain? Binding of ifenprodil does not stop the channel opening. Even at saturating concentrations of ifenprodil, the ion channel can still open, although with reduced duration of channel openings.

Response: The terminology for 'active', 'deactivated' and the 'ifenprodil-bound' state was taken from the different publications for the cryo-EM and crystal structures 5FXG, 5FXI and 4PE5, which were used to generate the homology models.^{7,8} These publications are also cited in the manuscript. For the active conformation the authors *Tajima et al.* showed, that the structure of 5FXG is connected to an open ion channel state. Since homology models were generated using only these structures, we used the same terminology. Ifenprodil is an allosteric modulator of the NMDAR and does not directly interact with the ion channel pore. Moreover, the inhibition of the NMDA receptor by ifenprodil and related compounds is state-dependent and restricts transitions to open state.⁹ Our findings and the published structures 5FXG clearly show, that binding of these compounds is only possible in the deactivated conformation. Even at saturating conditions an equilibrium between the ifenprodil-bound and ifenprodil-unbound state occurs. If ifenprodil is unbound while both agonists are bound the receptor will open and cannot be inhibited by ifenprodil until the receptor goes back to the deactivated conformation.

Minor points:

Page 12, IZ4 paragraph: insert 'an' before 'important' in line 7 of this paragraph.

Response: corrected.

What is the difference between SE and SEM in Table 1?

Response: SE is the standard error while SEM is the standard error of the mean. The SEM is calculated by the standard deviation from the mean of defined, single data points divided by \sqrt{n} . The standard error (SE) is used for derived parameters of an estimation function like dose response curves. In contrast to the SEM, the SE is calculated by using all raw data points and the other derived parameters of the dose response curve to calculate the error by an approximate formula (Error Propagation formula).

Was the pH used for the binding assay the same as that used for the TEVC experiments?

Response: The experiments were performed at almost identical pH values. TEVC experiments were performed at a pH value of 7.4 while the binding assay was performed at a pH value of 7.5. Both values are given in the material and methods section.

Legend to Figure-9, line 2: change 'vertical' to 'vertically'

Response: corrected.

Page-6, line 19: insert 'the' before 'following'

Response: corrected.

This paper is on an interesting topic but it is not really giving any advance in understanding compared to the structural work of the Gouaux group and the functional studies of the Traynelis group.

Response: We clearly disagree with this statement. First, with the class of 3-benzazepines we identified a new generation of ifenprodil-derived compounds, which are promising candidates for the development of neuroprotective agents. In particular, we characterized the importance of the phenolic and benzylic OH group and the flexibility of the phenylbutyl side chain for high inhibitory activity. Moreover, with compound (*R*)-**1** we identified a compound with a superior selectivity profile and higher activity compared to ifenprodil. Second, we identified for the first time the amino acid GluN2B F176 as the most critical interaction partner for ifenprodil and the compound class of 3-benzazepines. Together with the results for the amino acids GluN2B F114 and GluN2B Q110 we completed the picture of the corresponding interaction partners on the protein side, which were postulated by the well-established pharmacophore model. Third, we postulate an inhibition mechanism, which is in accordance with our results and with the structural work and functional studies of the two mentioned groups and other groups like the Furukawa group. Moreover, this mechanism of action explains the inhibitory activity on a small molecular scale, which was not achieved before. Additionally, since all known ifenprodil-derived compounds like CP-101,606, Ro 25-6981, eliprodil, EVT-101, the compounds from the 93-series and the 3-benzazepines clearly interact with GluN2B F176, we identified the general basis for inhibitory activity of all these GluN2B subunit-selective inhibitors.

References

- (1) Yuan, H.; Hansen, K. B.; Vance, K. M.; Ogden, K. K.; Traynelis, S. F. Control of NMDA Receptor Function by the NR2 Subunit Amino-Terminal Domain. *J. Neurosci.* **2009**, *29* (39), 12045–12058.
- (2) Tewes, B.; Frehland, B.; Schepmann, D.; Schmidtke, K.-U.; Winckler, T.; Wünsch, B. Conformationally Constrained NR2B Selective NMDA Receptor Antagonists Derived from Ifenprodil: Synthesis and Biological Evaluation of Tetrahydro-3-Benzazepine-1,7-Diols. *Bioorg. Med. Chem.* **2010**, *18* (22), 8005–8015.
- (3) Tewes, B.; Frehland, B.; Schepmann, D.; Schmidtke, K.-U.; Winckler, T.; Wünsch, B. Design, Synthesis, and Biological Evaluation of 3-Benzazepin-1-Ols as NR2B-Selective NMDA Receptor Antagonists. *ChemMedChem* **2010**, *5* (5), 687–695.
- (4) Dey, S.; Temme, L.; Schreiber, J. A.; Schepmann, D.; Frehland, B.; Lehmkuhl, K.; Strutz-Seeböhm, N.; Seeböhm, G.; Wünsch, B. Deconstruction - Reconstruction Approach to Analyze the Essential Structural Elements of Tetrahydro-3-Benzazepine-Based Antagonists of GluN2B Subunit Containing NMDA Receptors. *Eur. J. Med. Chem.* **2017**, *138*, 552–564.
- (5) Tewes, B.; Frehland, B.; Schepmann, D.; Robaa, D.; Uengwetwanit, T.; Gaube, F.; Winckler, T.; Sippl, W.; Wünsch, B. Enantiomerically Pure 2-Methyltetrahydro-3-Benzazepin-1-Ols Selectively Blocking GluN2B Subunit Containing N-Methyl-D-Aspartate Receptors. *J. Med. Chem.* **2015**, *58* (15), 6293–6305.
- (6) Yuan, H.; Myers, S. J.; Wells, G.; Nicholson, K. L.; Swanger, S. A.; Lyuboslavsky, P.; Tahirovic, Y. A.; Menaldino, D. S.; Ganesh, T.; Wilson, L. J.; Liotta, D. C.; Snyder, J. P.; Traynelis, S. F. Context-Dependent GluN2B-Selective Inhibitors of NMDA Receptor Function Are Neuroprotective with Minimal Side Effects. *Neuron* **2015**, *85* (6), 1305–1318.
- (7) Tajima, N.; Karakas, E.; Grant, T.; Simorowski, N.; Diaz-Avalos, R.; Grigorieff, N.; Furukawa, H. Activation of NMDA Receptors and the Mechanism of Inhibition by Ifenprodil. *Nature* **2016**, *534* (7605), 63–68.

- (8) Karakas, E.; Furukawa, H. Crystal Structure of a Heterotetrameric NMDA Receptor Ion Channel. *Science* (80-.). **2014**, *344* (6187), 992–997.
- (9) Bhatt, J. M.; Prakash, A.; Suryavanshi, P. S.; Dravid, S. M. Effect of Ifenprodil on GluN1/GluN2B N-Methyl-D-Aspartate Receptor Gating. *Mol. Pharmacol.* **2013**, *83* (1), 9–21.

REVIEWERS' COMMENTS:

Reviewer #1 (Remarks to the Author):

The authors carried out experiments and demonstrated that at 10uM, (R)-1 and (S)-1 showed similar selectivity as ifenprodil on GluN1/2B over 2A, 2C, and 2D NMDA receptors. Further, application of ifenprodil or (R)-1 inhibited evoked EPSC in hippocampal pyramidal cells.

The manuscript has been much improved. I have just a minor comment.

Did the authors use APV to confirm the recorded currents in whole-cell patch-clamp recordings are mediated by NMDA receptors? Why 6uM ifenprodil and 7uM (R)-1 were used during patch-clamp recording instead of 10uM? The inhibitory effect should be compared with previously reported findings (Kirson 1996, Tovar 1999, bellone 20017).

Reviewer #3 (Remarks to the Author):

COMMSBIO-19-0326A

A common mechanism allows selective targeting of GluN2B subunit-containing N-methyl-D-aspartate receptors

The authors have made a number of revisions to the manuscript that have improved the clarity of the paper. I think there are some areas where the paper could still be improved.

i) Hansen et al, (2014) described the low efficacy of ifenprodil at inhibiting triheteromeric NMDA receptors. At present, this seems to be the most likely explanation for the disappointing clinical trials with compounds in this class. It seems important that the authors should consider triheteromeric N1/N2A/N2B receptors in their Introduction and Discussion.

ii) In their rebuttal, page-2, the authors state, 'Ifenprodil and related compounds show only weak agonist dependency.' This is not the case. The effect has been widely studied, especially by Kew & Kemp. For example, their paper in Journal of Physiology in 1996: in the summary of the paper they state, 'ifenprodil binding to the NMDA receptor results in a 6-fold higher affinity for glutamate site agonists'.

iii) while the authors may disagree with my interpretation of the mechanism of ifenprodil action, the experimental evidence available in the literature strongly supports the idea that the NMDA receptor channel can open while ifenprodil is bound.

iv) Page-4, line-28: 'Electrophysiological studies and subsequent mathematical state modelling of the binding process suggest an induced-fit mechanism, which locks the receptor in closed state'. I suggest it would be better to not use the phrase, 'locks the receptor in the closed state'.

v) Page-11, line-19, 'transitions into the deactivated or ifenprodil-bound states'. I suggest it would be more correct to delete 'or' from this statement. Otherwise, the sentence is suggesting that ifenprodil does not bind to deactivated states.